

# Implementing the Water, HEat and Transport model in GEOframe WHETGEO-1D v.1.0: algorithms, informatics, design patterns, open science features, and 1D deployment.

Niccolò Tubini[1] and Riccardo Rigon[2]

[1]Department of Civil, Environmental and Mechanical Engineering, University of Trento, Via Mesiano 77, 38123 Trento, Italy
[2]Center Agriculture Food Environment, University of Trento, Via Mesiano 77, 38123 Trento, Italy

**Correspondence:** Niccolò Tubini (niccolo.tubini@unitn.it)

**Abstract.** This paper presents WHETGEO and its 1D deployment, a new, physically based model simulating the water and energy budgets in a soil column. The purpose of this contribution is twofold. First, we discuss the mathematical and numerical issues involved in solving the Richardson-Richards equation, conventionally known as Richards' equation, and the heat equation in heterogeneous soils. In particular, for the Richardson-Richards equation ($R^2$) we take advantage of the nested Newton-Casulli-Zanolli (NCZ) algorithm that ensures the convergence of the numerical solution in any condition. Second, starting from numerical and modelling needs, we present the design of a software that is intended to be the first building block of a new, customisable, land-surface model that is integrated with process-based hydrology. WHETGEO is developed as an open-source code, adopting the Object-Oriented paradigm and a generic programming approach in order to improve its usability and expandability. WHETGEO is fully integrated in the GEOframe/OMS3 system allowing the use of the many ancillary tools it provides. Finally the paper presents the 1D deployment of WHETGEO, WHETGEO-1D, which has been tested against the available analytical solutions presented in Appendix.

## 1  Introduction

The Earth's Critical Zone (CZ) is defined as the heterogeneous, near surface environment in which complex interactions involving rock, soil, water, air, and living organism regulate the natural habitat and determine the availability of life-sustaining resources (National Research Council, 2001). Clear interest in studying the CZ is spurred on by ever-increasing pressure due to the growth in human population and climatic changes. Central to simulating the processes in the CZ is the study of soil moisture dynamics (Clark et al., 2015a). In the following we suggest that studying the CZ requires tools that are not yet readily available to researchers; then we propose one of our own. These tools should be flexible enough to allow the quick embedding of advancements in science.





## 1.1 Setting Up the Water Budget

It is generally accepted in the field that the motion of water in soil is regulated by a Darcian-Stokesian flow, meaning that any force is immediately dissipated and water under a gradient of generalised forces acquires a velocity (the Darcy velocity) which is linearly proportional to the gradient of the generalised force. This law is known as the Darcy-Buckingham law and reads

$$\boldsymbol{J} = K(\theta)\boldsymbol{\nabla}\left(\psi + z\right) \tag{1}$$

where the forces acting are gravity ($z$ L) and the matric potential $\psi$ L, and where: $J$ LT$^{-1}$ is the Darcian flux; $K$ LT$^{-1}$ is the hydraulic conductivity; $\theta$ L$^3$L$^{-3}$ is the dimensioneless volumetric water content; $\boldsymbol{\nabla}$ L$^{-1}$ is the gradient operator; and $z$ L is the vertical coordinate, positive upward. The assumptions under which such a law is derived from Newton's law are presented in (Whitaker, 1986; Di Nucci, 2014). The hydraulic conductivity, $K$ LT$^{-1}$, depends on soil type (texture and structure) and water content, while the thermodynamic forces must be understood as gradients of the chemical potential of water, which, in turn, can be split as matric potential, osmotic potential and other potentials (Nobel et al., 1999, Chapter 2). However, in Eq. (1) we consider only the action of the matric potential. On the basis of the law of motion in Eq. (1), the mass conservation reads:

$$\frac{\partial \theta}{\partial t} = \nabla \cdot \left(K(\theta)\boldsymbol{\nabla}\left(\psi + z\right)\right) \tag{2}$$

where $\nabla\cdot$ L$^{-1}$ is the divergence operator. Equation (2) is usually known as the Richards' equation (Richards, 1931), but was previously formulated by Richardson (1922). Therefore, in the following we call it the R$^2$ equation to remind of this double origin. There are very informative reviews that cover its general, historical and numerical aspects, such as: (Paniconi and Putti, 2015; Farthing and Ogden, 2017; Zha et al., 2019). Therefore, it is not deemed necessary to further summarize the matter here. The R$^2$ equation is a function of two variables, $\theta$ and $\psi$, and its resolution requires another relation between these two quantities. This relation is known as soil-water retention curves (SWRC), written in the plural because we have many SWRC depending on soil characteristics. The reader may be aware that the R$^2$ is an exact description of unsaturated flow only if we assume that soil is a bundle of capillaries and that largest capillaries drain first and are filled last (Mualem, 1976). In fact, in this case a relation can be obtained between the radius of the capillaries and the suction, which was fully derived (Kosugi, 1999). However, there are various reasons to take the capillary bundle concept as a rough approximation of natural soil. To enumerate some of the issues:

1. Firstly, pores in soils are not bundles of well defined capillaries of a single diameter; in fact, they can have quite random structures, as revealed, for instance, by tomography (Yang et al., 2018).

2. Secondly, logic and pore scale simulations, as in Tomin and Lunati (2016) for example, indicate that fluids fill the cavities where they fall, and only eventually are they redistributed according to the microstructure of the soil; that is to say, fluids do not move instantaneously from the largest pores to the smallest ones.

3. A set of relatively large pores can, in certain conditions, preferentially drive the flow of water in a short time scale according to laminar viscous flow driven by gravity, before any redistribution happens (Germann and Beven, 1981).



4. The role of living matter, such as bacteria, animals, fungi, vegetation, and roots, is usually eliminated from the hydro-logical picture but it should have a relevant place (Benard et al., 2019).

Besides,

5. Capillary forces are not the only ones acting at the microscale (Lu, 2016). In fact, measured suction values are far below pressures that can be sustained by capillarity alone.

6. Temperature affects water viscosity; infiltration is faster at warm temperatures and slower at cold ones (Constantz and Murphy, 1991).

7. In high-latitude and high-elevation environments, soils may be subject to freezing and thawing processes which affect
pore volume and water dynamics (Dall'Amico et al., 2011).

These facts certainly do not threaten the nature of mass conservation in Eq. (2). However, they can certainly alter the statistics which generate the closure equations, i.e. the SWRC we currently use.

– Requirement I - Without entering into further details, we can observe that the aforementioned issues have consequences that would require a new software to include the possibility of adopting new parameterizations of SWRC and hydraulic
conductivity quickly, easily and neatly.

## 1.2 The Three or Four Worlds

The flow of water obeys the general laws of physics for conservation of mass and momentum but, since the seminal works of Freeze and Harlan in 1969, the scientific community has split up (Furman, 2008) into three groups: groundwater people, vadose zone scientists, and surface water hydrologists. This compartmentalisation of the scientific community was fostered
to deepen the knowledge within single branches, with the interactions between the different parts have been governed in models by assigning boundary conditions (Furman, 2008). However, these boundary conditions are intrinsically inadequate and inappropriate in representing the physics of interactions between different domains whose interactions depend strictly on the state of the system. When these conditions are prescribed a priori (Furman, 2008), the proper dynamics of the CZ fluxes cannot be obtained. There is the need to overcome this situation and, therefore:

– Requirement II - the boundary conditions hard-wired into algorithm implementation should be removed in favour of a simultaneous treatment of the three compartments (surface waters, vadose zone and groundwater).

Fortunately, Simunek et al. (2012) found the way to smoothly extend Richards equation into the groundwater equation. This and other similar approaches are now used in various codes, such as Hydrus, ParFlow (Ashby and Falgout, 1996; Jones and Woodward, 2001; Kollet and Maxwell, 2006), CATHY (Paniconi and Wood, 1993; Paniconi and Putti, 1994), and GEOtop 2.0
(Rigon et al., 2006a; Endrizzi et al., 2014). To extend the $R^2$ equation into the saturated domain it is necessary to include the contribution of groundwater storativity due matrix and fluid compressibility. The common approach is to write the $R^2$ equation





as:

$$\frac{\partial \theta}{\partial t} + S_s \frac{\theta}{\theta_s} \frac{\partial \theta}{\partial t} = \nabla \cdot (K(\psi)\boldsymbol{\nabla}(\psi + z)) \tag{3}$$

where $S_s\ \mathrm{L}^{-1}$ is the specific storage coefficient, defined as

$$S_s = \rho g(n\beta + \alpha), \tag{4}$$

with $\rho\ \mathrm{ML}^{-3}$ being the water density, $g\ \mathrm{LT}^{-2}$ gravitational acceleration, $n\ \mathrm{L}^3\mathrm{L}^{-3}$ the soil porosity, $\beta\ \mathrm{LT}^2\mathrm{M}^{-1}$ the liquid compressibility, and $\alpha\ \mathrm{LT}^2\mathrm{M}^{-1}$ the matrix compressibility. In the left-hand-side of Eq. (3), the first term accounts for changes in liquid saturation, while the second term accounts for the compression or expansion of the porous medium and the water. The left-hand-side term in Eq. (3) can be rewritten as

$$\left(c + S_s \frac{\theta}{\theta_s}\right)\frac{\partial \psi}{\partial t} \tag{5}$$

where $c\ \mathrm{L}^{-1}$ is the water retention capacity. Comparing the two terms in brackets, we can see that for $\psi < 0$, then $c >> S_s \frac{\theta}{\theta_s}$; this means that under unsaturated conditions, the contribution of the specific storage is negligible. Whereas when the soil is saturated and $\psi > 0$, then $c = 0$ and therefore what counts is the specific storage. Because of this, it is possible to account for groundwater specific storage simply by modifying the SWRC as:

$$\theta(\psi) = \begin{cases} \theta(\psi) & \text{if } \psi < 0 \\ \theta_s + S_s\psi & \text{if } \psi > 0 \end{cases} \tag{6}$$

Furthermore, switching from Richards to shallow water was made possible in the equation writing thanks to, for example, (Casulli, 2017; Gugole et al., 2018). Therefore, switching to a fully integrated, simultaneous treatment of the three domains is now possible.

### 1.3 The Necessary Coupling with the Energy Budget

As remarked in point 6 above, temperature affects water viscosity, which effectively doubles in passing from 5 to 20 degrees Celsius (Eisenberg et al., 2005), with a positive feedback on the infiltration process. This has been clearly observed in natural systems (Ronan et al., 1998; Eisenberg et al., 2005; Engeler et al., 2011) where infiltration rates follow diurnal and seasonal temperature-cycles. In fact, according to Muskat and Meres (1936), the unsaturated hydraulic conductivity can be expressed as

$$K(\theta) = \kappa_r(\theta)\,\kappa\frac{\rho\,g}{\nu} \tag{7}$$

where $\kappa_r(\theta)-$ is the relative permeability, $\kappa\ \mathrm{L}^2$ is the intrinsic permeability, $\rho\ \mathrm{L}^3\mathrm{M}^{-1}$ is the liquid density, $g$ is the acceleration of gravity, and $\nu\ \mathrm{L}^2\mathrm{T}^{-1}$ is the kinematic viscosity of the liquid. Thus, for constant $\theta$, variations in $K(\theta)$ due to temperature can be accounted as (Constantz and Murphy, 1991):

$$K(\theta, T_2) = K(\theta, T_1)\frac{\nu(T_1)}{\nu(T_2)} \tag{8}$$





Temperature is also responsible for the phase change of water, point 7, and because of pore ice, as well as excess ice, infiltration rates and subsurface flows are significantly modified Walvoord et al. (2012).

  – Requirement III - To account for thermodynamic effects, temperature should be at least present in the $R^2$ equation as a parameter, as in Eq. (8). However, for a more accurate approximation of the water dynamics, the option to solve the water and energy budgets simultaneously must be present.

Soil thermal properties are important physical parameters in modelling land surface processes (Dai et al., 2019) since they control the partitioning of energy at the soil surface and its redistribution within the soil (Ochsner et al., 2001). For a multi-phase material, like soil, their definition is always problematic since they depend on the physical properties of each phase and their variations (Dong et al., 2015; Dai et al., 2019; Nicolsky and Romanovsky, 2018). In literature different models have been proposed with such a scope, and further studies on it are recommended (Dai et al., 2019): nonetheless, when considering the

phase change of water, the estimation of the unfrozen and frozen water fraction is still an unresolved issue for which different models, usually referred to as SFCC, Soil Freezing Characteristics Curve, have been proposed (Kurylyk and Watanabe, 2013). Thus, it is clear that the aspects related to the estimation of soil thermal properties fall fully within Requirement I too. Moreover, there are other reasons for which the equations of the water and energy budgets should be solved in a coupled manner (Rigon et al., 2006a): for instance, this makes it possible to include an appropriate treatment of evaporation and transpiration processes

(Bonan, 2019; Bisht and Riley, 2019), as well as of the heat advection (Frampton et al., 2013; Walvoord and Kurylyk, 2016; Wierenga et al., 1970; Ronan et al., 1998; Engeler et al., 2011; Zhang et al., 2019).

  Finally, there is a great urge to model solutes transport according to the water movements. The range of applications for solute/tracer/pollutants spans from agriculture to industry to research itself. In fact, in recent years there has been a tumultuous growth of studies using tracers to asses the various pathways of water (Hrachowitz et al., 2016). However, so far these studies

have mostly used lumped models with limited capability to investigate water age selection processes, processes that became very important in the most recent literature e.g. (Penna et al., 2018). Using more complex modelling can benefit both the investigation of the processes and the construction of more refined water budget closures. Even though in this paper we do not detail the work on tracers, they must be kept in mind in software design so that the modules to be implemented eventually.

### 1.4 Heat Transport

Under the conditions defined above, the governing equation for the transport of energy in variably saturated porous media is given by the following energy conservation equation:

$$\frac{\partial h(\psi, T)}{\partial t} = \nabla \cdot [\lambda(\psi)\nabla T - \rho_w c_w(T - T_{ref})J_w] \tag{9}$$

where $h$ is the specific enthalpy of the medium $L^2T^{-2}$, $\lambda$ $ML\Theta^{-1}T^{-3}$ is the thermal conductivity of the soil, $T$ $\Theta$ is the temperature, $\rho_w$ $ML^{-3}$ is the water density, $c_w$ $L^2\Theta^{-1}T^{-2}$ is the specific heat capacity of water, $T_{ref}$ $\Theta$ is a reference

temperature used to define the enthalpy, and $J_w$ is the water flux $LT^{-1}$. The first term on the right-hand-side is the heat conduction flux, described by the Fourier's law, and the second term is the sensible heat advection of liquid water. The specific


enthalpy of a control volume of soil $V_c$ L$^3$ can be calculated as the sum of the enthalpy of the soil particles and liquid water:

$$h = \rho_{sp} c_{sp} (1 - \theta_s)(T - T_{ref}) + \rho_w c_w \theta(\psi)(T - T_{ref}) \tag{10}$$

where $\rho_{sp}$ and $\rho_w$ are the densities of the soil particles and the water, $c_{sp}$ and $c_w$ are the specific heat capacities of the soil
particles and the water. Equation (9) is the so-called conservative form.

### 1.5 Organisation and Scope

This paper describes the implementation and content of the WHETGEO-1D (Water, HEat and Transport in GEOframe) software, in observance of requirements I to III and aware of the hydrologic facts described in points 1 to 7 of Section 1.1. Further requirements derive from Numerics and Mathematics, as argued below. We do not treat transport here as its numerics is slightly
different from those of the water and energy budgets;this topic will be covered in dedicated paper.

GEOframe (Formetta et al., 2014a; Bancheri, 2017; Bancheri et al., 2020) is a system of components, see Appendix A, built upon the Object Modelling System v3 (OMS3) framework (David et al., 2013), see Appendix B. GEOframe provides various components for precipitation treatment, radiation estimation in complex terrain, and evaporation and transpiration, that can be connected to WHETGEO-1D to create different modelling solutions. Because of the modularity of OMS3, WHETGEO-
1D components can be seamlessly used at run time by connecting them with the OMS3 DSL language based on Groovy (https://groovy-lang.org). OMS3 provides the basic services and, among them, tools for calibration and implicit parallelization of component runs. With WHETGEO-1D the GEOframe system is enriched with new components, and it is able to cope with different spatial resolutions and configurations compared to what was offered by existing components (Bancheri et al., 2020). Moreover, WHETGEO-1D represents the first brick of a new expandable land surface model (Bisht and Riley, 2019):
the forthcoming development of WHETGEO-1D is the Lysimeter GEO model that aims to simulate the soil-plant continuum (D'Amato et al., 2022).

In the following section we discuss the mathematical issues and the numerics of our implementation. In the subsequent one, we describe the informatics and software engineering issues, and how we solved them. Finally, we discuss WHETGEO-1D implementation by testing it against some analytical solutions available for simplified parameterizations and boundary
conditions, for details of these see Appendix C.

## 2 Mathematical Issues and Numerics

Translating these equations into numerical discretized codes implies to overcome some challenges, as we illustrate in the next sections by exploring, at first, the issues posed by each one of the equation.

### 2.1 General Issues of the R$^2$ Equation

Equation (2) is said to be written in "mixed form" because it is expressed in term of $\theta$ and $\psi$ (and uses the SWRC to connect the two variables).





The "$\psi$-based form" is derived from Eq. (2) by applying the chain rule for derivatives:

$$c(\psi)\frac{\partial \psi}{\partial t} = \nabla \cdot [K(\psi)\nabla(\psi + z)] \tag{11}$$

where

$$c(\psi) = \frac{\partial \theta(\psi)}{\partial \psi} \tag{12}$$

with dimension $\mathrm{L}^{-1}$, is the specific moisture capacity, also called hydraulic capacity. Even though Eq. (2) and Eq. (11) are analytically equivalent under the assumption that the water content is a differentiable variable, this is not generally true in the discrete domain where the derivative chain rule is not always valid (Farthing and Ogden, 2017). Because of this the $\psi$-based form may suffer from large balance errors in the presence of big nonlinearities and strong gradients, as discussed in (Casulli and Zanolli, 2010; Farthing and Ogden, 2017; Zha et al., 2019) and literature therein. The specific moisture capacity $c$, which appears in the storage term, itself depends on $\psi$ and so is not constant over a discrete time interval during which $\psi$ changes value. Let us discretise the time derivative in Eq. (11) by using the backward Euler scheme and obtain:

$$\tilde{c}_i \frac{\psi_i^{n+1} - \psi_i^n}{\Delta t} \tag{13}$$

where $\tilde{c}_i$ is the discrete operator of the soil moisture capacity, $c(\psi)$. In order to preserve the chain rule of derivatives at the discrete level, i.e. the equality $c\partial\psi/\partial t = \partial\theta(\psi)/\partial t$, $\tilde{c}_i$ has to satisfy the requirement (Roe, 1981)

$$\tilde{c}_i(\psi_i^{n+1} - \psi_i^n) = \theta(\psi_i^{n+1}) - \theta(\psi_i^n) \tag{14}$$

As can be seen from the above equation, the right definition of $\tilde{c}_i$ depends on the solution itself. To overcome this problem, in literature different techniques have been presented to improve the evaluation of $\tilde{c}_i$, but none ensures mass conservation (Farthing and Ogden, 2017).

There is a third form of Eq. (2), the so-called "$\theta$-based form", that reads as

$$\frac{\partial \theta}{\partial t} = \nabla \cdot [D(\theta)\nabla\theta + K(\theta)] \tag{15}$$

where all is made explicit by inverting the SWRC and $D(\theta) := K(\theta)\partial\psi/\partial\theta$ is the soil-water diffusivity $\mathrm{L}^2\mathrm{T}^{-1}$. The first term on the right-hand-side represents the water flow due to capillary forces, while the second term is the contribution due to gravity (Farthing and Ogden, 2017). The $\theta$-based form is mass-conserving and it can be solved perfectly by mass conservative methods (Casulli and Zanolli, 2010). However, it applicability is limited to the unsaturated zone since water content varies between $\theta_r$ and $\theta_s$, whereas water suction is not bounded. This formulation is intrinsically not suited to fulfilling our requirement III. Moreover, water content is discontinuous across layered soil since the SWRCs are soil specific, whereas water suction is continuous even in inhomogeneous soils (Farthing and Ogden, 2017; Bonan, 2019).

In WHETGEO, we directly use the conservative form of the $\mathrm{R}^2$ equation, Eq. (2), which seems the easiest way to deal with both the mass conservation issues and the extension of the equation to the saturated case.





### 2.1.1 The Discretization of the R$^2$ Equation

The implicit finite volume discretization of Eq. (2) reads as:

$$\theta_i(\psi_i^{n+1}) = \theta_i(\psi_i^n) + \Delta t \left[ K_{i+\frac{1}{2}}^{n+1} \frac{\psi_{i+1}^{n+1} - \psi_i^{n+1}}{\Delta z_{i+\frac{1}{2}}} + K_{i+\frac{1}{2}}^{n+1} - K_{i-\frac{1}{2}}^{n+1} \frac{\psi_i^{n+1} - \psi_{i-1}^{n+1}}{\Delta z_{i-\frac{1}{2}}} - K_{i-\frac{1}{2}}^{n+1} + S_i^n \right] \tag{16}$$

where $\Delta t$ is the time step size,

$$S_i = \int_{\Omega_i} S \, d\Omega \tag{17}$$

is an optional source/sink term in volume, and $\theta_i(\psi)$ is the $i$th water volume given by

$$\theta_i(\psi) = \int_{\Omega_i} \theta(\psi) \, d\Omega. \tag{18}$$

Equation (16) can be written in matrix form as

$$\boldsymbol{\theta}(\boldsymbol{\psi}) + \mathbf{T}\boldsymbol{\psi} = \boldsymbol{b} \tag{19}$$

where $\boldsymbol{\psi} = \{\psi_i\}$ is the tuple of unknowns, $\boldsymbol{\theta}(\boldsymbol{\psi}) = \theta_i(\psi_i)$ is a tuple-function representing the discrete water volume, $\mathbf{T}$ is the flux matrix, and $\boldsymbol{b}$ is the right-hand-side vector of Eq. (16), which is properly augmented by the known Dirichlet boundary condition when necessary. For a given initial condition $\psi_i^0$, at any time step $n = 1, 2, \ldots$, Eq. (16) constitutes a nonlinear system for $\psi_i^{n+1}$, with the nonlinearity affecting only the diagonal of the system and being represented by the water volume $\theta_i(\psi_i^{n+1})$. This set of equations is a consistent and conservative discretization of Eq. (2). Therefore, regardless of the chosen spatial and temporal resolution, $\psi_i^{n+1}$ is a conservative approximation of the new water suction.

### 2.1.2 Surface Boundary Condition

The definition of the type of surface boundary condition (Neumann vs. Dirchlet) for the R$^2$ equation is a non trivial task since it can depend on the state of the system. In literature several approaches are used (Furman, 2008). These approaches are mainly based on a switch of the type of the boundary condition from a prescribed head to prescribed flux and viceversa. This switching often causes numerical difficulties that need to be addressed (Furman, 2008).

To overcome this problem we have included an additional computational node at the soil surface. As will be made clear in the following, for it we prescribe an "equation state" like that presented in (Casulli, 2009):

$$H(\psi) = \begin{cases} \psi & \text{if } \psi > 0 \\ 0 & \text{otherwise} \end{cases} \tag{20}$$

where $H$ L is the water depth, which also represents the pressure if the ponding is assumed to happen in hydrostatic conditions. By doing so, it is possible to prescribe as the surface boundary condition the rainfall intensity (Neumann type) without resorting




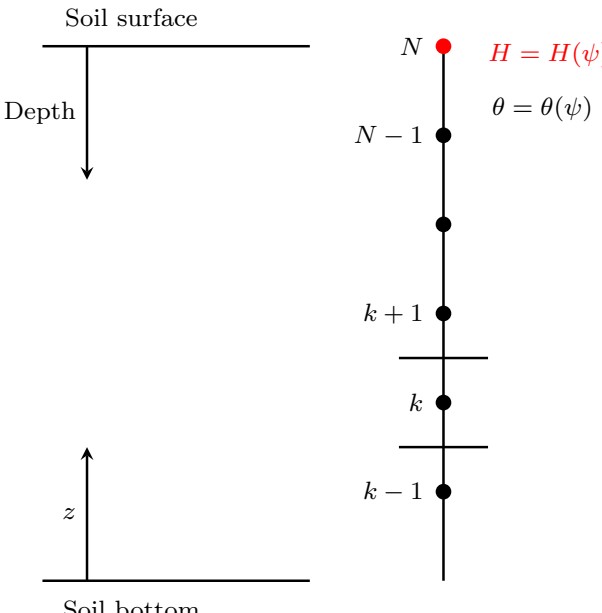

**Figure 1.** Scheme of the computational domain to solve the Richards equation in 1D. The uppermost node represents the water depth at the soil surface. By considering this additional computational node the boundary condition does not change its nature depending on the solution.

to any switching techniques to reproduce infiltration excess or saturation excess processes. In this case the system, Eq. (19), must to be modified to account for the additional computational node describing the state of the soil surface:

$$
\begin{cases}
H_N(\psi_N^{n+1}) - \Delta t \left[ 0 - K_{N-\frac{1}{2}}^n \dfrac{\psi_N^{n+1} - \psi_{N-1}^{n+1}}{\Delta z} \right] = H_N(\psi_N^n) + \Delta t[J^n - K_{N-\frac{1}{2}}^n] & \text{if } i == N \\
\theta_i(\psi_i^{n+1}) - \Delta t \left[ K_{i+\frac{1}{2}}^n \dfrac{\psi_{i+1}^{n+1} - \psi_i^{n+1}}{\Delta z} - K_{i-\frac{1}{2}}^n \dfrac{\psi_i^{n+1} - \psi_{i-1}^{n+1}}{\Delta z} \right] = \theta_i(\psi_i^n) + \Delta t[K_{i+\frac{1}{2}}^n - K_{i-\frac{1}{2}}^n] & \text{if } i = 1, 2, ..., N-1
\end{cases}
\tag{21}
$$

where $J^n$ is the rainfall intensity and represents the Neumann boundary condition used to drive the system at the soil surface.

For any time step, Eq. (21) can be written in matrix form, similar to (Eq. (19)), as:

$$
\boldsymbol{V}(\boldsymbol{\psi}) + \mathbf{T}\boldsymbol{\psi} = \boldsymbol{b}
\tag{22}
$$

where $\boldsymbol{\psi} = \{\psi_i\}$ is the tuple of unknowns, $\boldsymbol{V}(\boldsymbol{\psi}) = (\theta_i(\psi_i))$ for $i = 1, 2, ..., N-1$ and $V_N(\psi) = H(\psi)$ is a tuple-function representing the discrete water volume, $\mathbf{T}$ is the flux matrix, and $\boldsymbol{b}$ is the right-hand-side vector of Eq. (16), which is properly augmented by the known Dirichlet boundary condition when necessary. For a given initial condition $\psi_i^0$, at any time step

$n = 1, 2, ...$ Eq. (22) constitutes a nonlinear system for $\psi_i^{n+1}$, with the nonlinearity affecting only the diagonal of the system and being represented by the water volume $V_i(\psi_i^{n+1})$. Therefore, regardless of the chosen spatial and temporal resolution, $\psi_i^{n+1}$ is a conservative approximation of the new water suction.





## 2.2 Heat Transport Numerics Issues

Equation (9) is said to be written in conservative form and expresses an important property, which is the conservation of the
scalar quantity, in this case the specific enthalpy. It is interesting to note that by making use of the mass conservation equation,
Eq. (2), Eq. (9) can be written in an analytically equivalent form, the so called non-conservative form (Sophocleous, 1979;
Simunek et al., 2005):

$$c_T \frac{\partial T}{\partial t} = \lambda \nabla^2 T - \rho_w c_w J_w \nabla T \tag{23}$$

Equation (23) expresses another important property which is the maximum principle (Casulli and Zanolli, 2005), i.e. the
analytical solution is always bounded, above and below, by the maximum and minimum of its initial and boundary values, as
shown in (Greenspan and Casulli, 1988, Chapter 7.3).

Although Eq. (9) and Eq. (23) are analytically equivalent, once they are discretised the corresponding numerical solution
will, in general, either be conservative or satisfy a discrete max-min property (Casulli and Zanolli, 2005), but not both as would
be required.

As in the case of the water flow, thermal budget issues can be subdivided into three aspects: the discretization of the equation,
the inclusion of the appropriate boundary conditions, and the implementation of some closure equation for the thermal capacity
and conductivity.

### 2.2.1 The Discretization of the Heat Equation

The key feature (Casulli and Zanolli, 2005) to obtaining a numerical solution for the heat transport equation that is both
conservative and possesses the max-min property is to solve the conservative form of the heat equation by using the velocity
field obtained in solving the continuity equation Eq. (2). By making use of the up-wind scheme for the advection part and the
centered difference scheme for the diffusion part we have:

$$C_{T_i}^{n+1} T_i^{n+1} = C_{T_i}^n T_i^n - \rho_w c_w \Delta t \left[ \frac{1}{2} J_{1+\frac{1}{2}}^{n+1} \left( T_{i+1}^{n+1} + T_i^{n+1} \right) - \frac{1}{2} \left| J_{1+\frac{1}{2}}^{n+1} \right| \left( T_{i+1}^{n+1} + T_i^{n+1} \right) \right.$$

$$\left. - \frac{1}{2} J_{1-\frac{1}{2}}^{n+1} \left( T_i^{n+1} + T_{i-1}^{n+1} \right) + \frac{1}{2} \left| J_{1-\frac{1}{2}}^{n+1} \right| \left( T_{i+1}^{n+1} + T_i^{n+1} \right) \right]$$

$$+ \Delta t \left[ \lambda_{i+\frac{1}{2}}^n \frac{T_{i+1}^{n+1} - T_i^{n+1}}{\Delta z} - \lambda_{i-\frac{1}{2}}^n \frac{T_i^{n+1} - T_{i-1}^{n+1}}{\Delta z} \right] \tag{24}$$

where

$$C_{T_i} = \int_{\Omega_i} \rho_{sp} c_{sp} (1 - \theta_s) + \rho_w c_w \theta(\psi) \, d\Omega \tag{25}$$

When heat equation does not consider water phase changes, it is decoupled from the $R^2$ equation and the finite volume discreti-
sation leads to a linear algebraic system of equations. However, once freezing and thawing processes are considered, the heat





equation is fully coupled with the R$^2$ equation, as in (Dall'Amico et al., 2011) for instance, and the enthalpy function becomes nonlinear. At this point, since the enthalpy function is nonlinear the NCZ algorithm is required to linearise it, as shown in (Tubini et al., 2020). So far, we have not considered the problem of water flow in freezing soils, however being aware of this issue is important for the future developments and code design.

**2.2.2 Driving the Heat Equation with the Surface Energy Budget**

At the soil surface the heat equation is driven by the surface energy balance. The heat flux exchanged between the soil and the atmosphere, the surface heat flux, $\mathcal{F}\ \mathrm{MT}^{-3}$, is given as:

$$\mathcal{F} = S_{in} + S_{out} + L_{in} + L_{out} + H + LE \tag{26}$$

where $S_{in}$ is the incoming short wave radiation, $S_{out}$ is the outgoing short wave radiation, $L_{in}$ is the incoming longwave
radiation, $L_{out}$ is the outgoing longwave radiation, and $H$ and $LE$ are respectively the turbulent fluxes of sensible heat and latent heat. Fluxes are positive when directed toward the soil surface and all have the dimension of an energy per unit area per unit time $\mathrm{MT}^{-3}$.

Similarly to the definition of the surface boundary condition for the water flow, the surface boundary condition for the energy equation is also system dependent. In fact, in (26) the only fluxes that do not depend on the soil temperature and/or moisture
are the incoming shortwave and longwave radiation fluxes, $S_{in}$ and $L_{in}$. The outgoing shortwave radiation flux is usually parameterized as:

$$S_{out} = \alpha S_{in} \tag{27}$$

where the surface albedo $\alpha-$ can be assumed to vary with the soil moisture content (Saito et al., 2006) and radiation wavelength. The outgoing longwave surface radiation is:

$$L_{out} = (\epsilon - 1)L_{in} - \epsilon \sigma T_s^4 \tag{28}$$

where $T_s\ \Theta$ the temperature of the topmost layer of soil, $\epsilon$ is the soil emissivity, and $\sigma$ is the Stefan-Boltzmann constant. The sensible heat flux $H$ is taken as:

$$H = -\frac{\rho_a c_a}{r_H}(T_a - T_s) \tag{29}$$

where $\rho_a$ is the air density $\mathrm{ML}^{-3}$, and $c_a$ is the thermal capacity of air per unit mass $\mathrm{L}^2\mathrm{T}^{-2}\Theta^{-1}$. Regarding the aerodynamic
resistances $r_H\ \mathrm{TL}^{-1}$, it should be noted that it can be evaluated with different degrees of approximation and may require a specific modelling solution. For instance, the aerodynamic resistance $r_H$ can be evaluated with models ranging from semi-empirical models to the the Monin-Obukov similarity (Liu et al., 2007), or even by solving the turbulent dynamics with direct methods (Raupach and Thom, 1981; Mcdonough, 2004).

The latent heat flux is taken here as given by a formula of the type:

$$LE = l\rho_a E_P \frac{r_H r_v}{r_H + r_v} \tag{30}$$





where $l$ $\mathrm{ML^2T^{-2}}$ is the specific latent heat of vaporisation of water, $E_P$ is the potential evapotranspiration, $r_H$ and $r_v$ $\mathrm{TL^{-1}}$ are respectively the aerodynamic resistance and the soil surface resistance to water vapour flow. The latent heat flux it is the sum of two distinct processes evaporation and transpiration. Compared to the other fluxes, latent heat flux presents further complications because evaporation is both an energy and a water limited process, and transpiration depends also on the physi-

ology of trees (as well as root distribution/growth and leaf cover). The latent heat flux is associated to the water flux that must be accounted in the $R^2$ equation. Here we present a simplified treatment of the latent heat flux as an external driving force/ prescribed boundary condition. A more exhaustive and physically based treatment of the latent heat flux, and the related water flux, is addressed in the ongoing development of the Lysimeter GEO model (D'Amato).

Including the surface energy budget boundary condition requires the computation of additional quantities such as the in-

coming radiation fluxes, the shortwave radiation and the longwave radiation, and the potential evapotranspiration flux. These quantities can be easily computed within the GEOframe system in which WHETGEO-1D is embedded. The proper estimation of the incoming radiation fluxes is far from being a simple task and it is often oversimplified in hydrological problems. Our approach is to use the tools already developed inside the system GEOframe which were tested independently and accurately (Formetta et al., 2013, 2014a). Similarly the evapotranspiration can be computed with other GEOframe components (Bottazzi,

2020; Bottazzi et al., 2021).

## 2.3 Algorithms

By using a numerical method, here the finite volume method, a partial differential equation is transformed into a system of nonlinear algebraic equations, as has already been shown. The system has to be solved with iterative methods and, at their core, these reduce the problem to using a linear systems solver. The solver can be of various types, according to the dimension of

the problem. For instance in 1D, the final system of finite volume problems we present is tridiagonal and can be conveniently solved with the Thomas algorithm (Quarteroni et al., 2010), which is a fast direct method. In 2D or 3D, the final matrix is not tridiagonal and a different solution method must be used, for instance the conjugate gradient (Shewchuk et al., 1994). These algorithms are well known and do not need to be explained here.

However, the reduction of a nonlinear system to a linear one is not trivial. We illustrate the issues by taking the $R^2$ equation as

an example. As discussed in depth in Zha et al. (2019) and Farthing and Ogden (2017), and references therein, the linearisation of the $R^2$ equation is challenging. Following the work of Celia et al. (1990), a lot of advancements have been made in this direction: Hydrus, CATHY, and ParFlow use variants of the Newton and Picard iteration methods (Zha et al., 2019; Paniconi and Putti, 1994), while GEOtop 2.0 implements a suitable globally convergent Newton method (Kelley, 2003). Although current algorithms are relatively stable, they may fail to converge or require a considerable computational cost (Zha et al.,

2019). This has a significant impacts both on the reliability of the solution, which can have mass-balance errors, and on the computational cost to produce it (Farthing and Ogden, 2017; Zha et al., 2019). Since Casulli and Zanolli (2010) and Brugnano and Casulli (2008), a new method was found, called nested Newton by the authors and NCZ in the following, that guarantees convergence in any situation, even with the use of large time steps and grid sizes.





As clearly pointed out by Casulli and Zanolli (2010), what makes the linearisation of the $R^2$ equation difficult is the non-
monothonic behaviour of the soil moisture capacity. A mathematical proof of convergence for NCZ exists (Brugnano and
Casulli, 2008, 2009; Casulli and Zanolli, 2010, 2012), which is not repeated here. However, we take the time to illustrate this
new algorithm with care.

Let us start again from the nonlinear system (Casulli and Zanolli, 2012):

$$\boldsymbol{V}(\boldsymbol{\psi}) + \mathbf{T}\boldsymbol{\psi} = \boldsymbol{b} \tag{20}$$

where $\boldsymbol{\psi} = (\psi_i)$ is the tuple of unknowns, $\boldsymbol{V}(\boldsymbol{\psi}) = (V_i(\psi_i))$ is a nonnegative vectorial function and where the $V_i(\psi_i)$ are
defined for all $\psi_i \in \mathbb{R}$ and can be expressed as:

$$V_i(\psi_i) = \int_{-\infty}^{\psi_i} a_i(\xi)\, d\xi \tag{31}$$

For all $i = 1, 2, ..., N$, the following assumptions are made on the functions $a_i(\psi)$ (we are here quite literally following (Casulli
and Zanolli, 2010)):

A1 : $a_i(\psi)$ is defined for all $\psi \in \mathbb{R}$ and is a nonnegative function with bounded variations;

     A2 : There exists $\psi_i^* \in \mathbb{R}$ such that $a_i(\psi)$ is strictly positive and nondecreasing in $(-\infty, \psi_i^*)$ and nonincreasing in $(\psi_i^*, +\infty)$.

$\mathbf{T}$ in eq (Eq. (20)) is the so-called matrix flux and it is a symmetric and (at least) positive semidefinite matrix satisfying one of
the following properties:

     T1 : $\mathbf{T}$ is a Stieltjes matrix, i.e., a symmetric M-matrix, or

T2 : $\mathbf{T}$ is irreducible, $\text{null}(\mathbf{T}) \equiv \text{span}(\boldsymbol{v})$, with $\boldsymbol{v} > \boldsymbol{0}$ (componentwise), and $\mathbf{T} + \mathbf{D}$ is a Stieltjes matrix for all diagonal
          matrices $\mathbf{D} \gneq \mathrm{O}$, with O denoting the null matrix.

Finally $\boldsymbol{b}$ is the vector of the known terms. When $\mathbf{T}$ satisfies property T2, then for Eq. (20) to be physically and mathematically
compatible, the following assumption about $\boldsymbol{b}$ is required:

$$0 < \boldsymbol{v}^\mathsf{T}\boldsymbol{b} < \boldsymbol{v}^\mathsf{T}\boldsymbol{V}^{Max} \tag{32}$$

where $\boldsymbol{V}^{Max} = \int_{-\infty}^{+\infty} a_i(\xi)d\xi$.

Having assumed that the $a_i(\psi)$ are non-negative functions of bounded variations, they are differentiable almost everywhere,
admit only discontinuities of the first kind, and can be expressed as the difference of two non-negative, bounded, and non-
decreasing functions, say $p_i(\psi)$ and $q_i(\psi)$, such that:

$$a_i(\psi) = p_i(\psi) - q_i(\psi) \geq 0 \tag{33}$$

$0 \leq q(\psi) \leq p(\psi)$

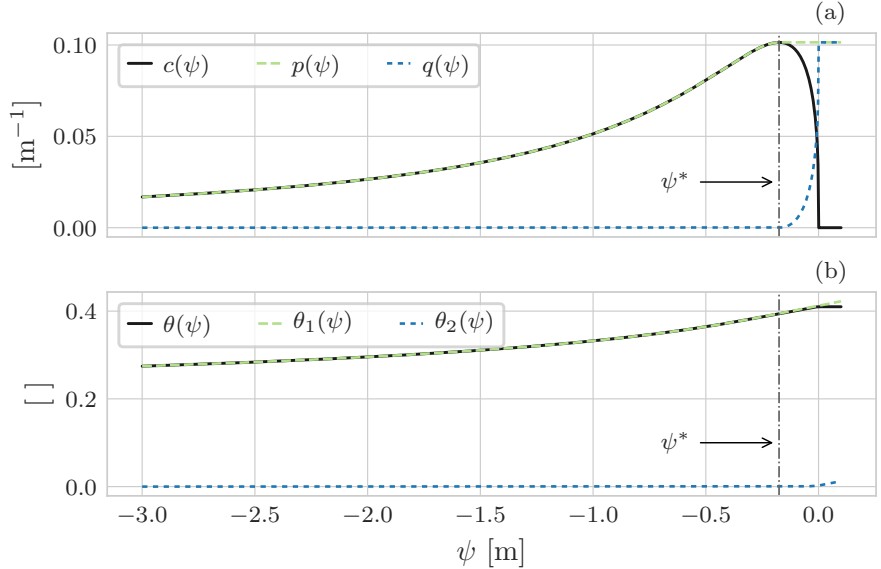

**Figure 2.** Graphical representation of the Jordan decomposition for soil water content using the SWRC model by Van Genuchten (Van Genuchten, 1980) for a clay loam soil (Bonan, 2019). (a) shows the Jordan decomposition of $c(\psi)$, (34). For $\psi = \psi^*$, $c(\psi)$ presents a maximum: for $\psi < \psi^*$ it is increasing, and for $\psi > \psi^*$ it is decreasing. This non monothonic behaviour causes problems when solving the nonlinear system. $c(\psi)$ is thus replaced by $p(\psi)$ (in green) and $q(\psi)$, in blue, two monothonic functions whose difference is the original function $c$. Consequently, (b), $\theta(\psi)$ is replaced by $\theta_1(\psi)$ and $\theta_2(\psi)$, Eq. (35).

for all $\psi \in \mathbb{R}$. When $a(\psi)$ satisfy assumptions A1 and A2, the corresponding decomposition (known as the Jordan decomposition (Chistyakov, 1997) and presented in Figure 2), is given by:

$$p_i(\psi) = a_i(\psi) \qquad\qquad q_i(\psi) = 0 \qquad\qquad\qquad\qquad \text{if } \psi \leq \psi_i^* \qquad (34)$$
$$p_i(\psi) = a_i(\psi_i^*) \qquad\qquad q_i(\psi) = p_i(\psi) - a_i(\psi) \qquad\qquad \text{if } \psi > \psi_i^*$$

where $\psi_i^*$ is the position of the maximum of $p_i$. Thereafter, $\boldsymbol{V}(\boldsymbol{\psi})$ can be expressed as

$$\boldsymbol{V}(\boldsymbol{\psi}) = \boldsymbol{V}_1(\boldsymbol{\psi}) - \boldsymbol{V}_2(\boldsymbol{\psi}) \qquad (35)$$

where the $i$-th component of $\boldsymbol{V}_1(\boldsymbol{\psi})$ and $\boldsymbol{V}_2(\boldsymbol{\psi})$ are defined as

$$V_{1,i}(\psi_i) = \int_{-\infty}^{\psi_i} p_i(\xi)\, d\xi \qquad V_{2,i}(\psi_i) = \int_{-\infty}^{\psi_i} q_i(\xi)\, d\xi \qquad (36)$$

By making use of Eq. (35) the algebraic system in Eq. (20) can be written as

$$\boldsymbol{V}_1(\boldsymbol{\psi}) - \boldsymbol{V}_2(\boldsymbol{\psi}) + \mathbf{T}\boldsymbol{\psi} = \boldsymbol{b} \qquad (37)$$





It is necessary here to point out exactly how the nonlinear system, Eq. (20), reads when considering only the $R^2$ equation and when the water depth function is used to properly define the surface boundary condition. In the first case, i.e. when Neumann or Dirichlet boundary conditions are used, the vectorial function is defined as $\boldsymbol{V}(\boldsymbol{\psi}) = (\theta_i(\psi_i))$ for $i = 1, 2, ..., N$.

Instead, when we consider the water depth function to describe the computational node at the soil surface, the vectorial function is defined as $\boldsymbol{V}(\boldsymbol{\psi}) = (\theta_i(\psi_i))$ for $i = 1, 2, ..., N-1$ and $V_N(\psi) = H(\psi)$. Therefore, the nonlinear system in Eq. (37) is valid to describe both the subsurface and surface waters when the symbols are appropriately understood.

This aspect, the use of two different equation states, and the fact that the NCZ algorithm can be successfully reused to solve other problems (Casulli and Zanolli, 2012; Tubini et al., 2020), requires a careful design of its implementation, as discussed in the following sections.

## 3  Informatics

The concepts and requirements previously illustrated must be cast into a software whose usability, expandability and inspectability are demanded by good software design, which adds further requirements. As discussed in Serafin (2019), codes were usually developed as monolithic code with severe drawbacks for maintainability and developments to improve the description of environmental processes, as has been proven by our own experience with the model GEOtop (Rigon et al., 2006a; Endrizzi et al., 2014), and by the experiences of other modelling frameworks (Clark et al., 2015b, 2021; Bisht and Riley, 2019). Based on these experiences, the WHETGEO-1D code has been developed by adopting an Object-Oriented-Programming (OOP) approach and it has been integrated into OMS3. Information on OMS3 is provided in Appendix B. Furthermore, WHETGEO-1D is part of the system of interoperable components called GEOframe, a short description of which is given in Appendix A. The utility of GEOframe has been partially discussed previously, when treating the surface energy budget.

### 3.1  Design Requirements

One of the major difficulties encountered by a research group concerns the development and reuse of scientific software (Berti, 2000) and the writing of structurally clean code, i.e. a code that is easily readable and understandable, with objects that have a specified, and possibly unique, responsibility (Martin, 2009).

An Object-Oriented-Programming approach, with the adoption of standard design patterns (DPs) (Gamma et al., 1995; Freeman et al., 2008) and the creation of new ones, has been adopted for the internal classes design and hierarchy.

The design principles followed by the WHETGEO-1D software can be summarised as follows:

A. The software should be open source to allow inspection and improvements by third parties;

B. For the same reason it should be organised into parts, each with a clear functional meaning and possibly a single responsibility.

C. The software can be extended with minimal effort and without modifications (according to the "open to extensions, closed to modifications" principle Freeman et al. (2008)). In particular, the parts to be modified are those that, according





to the discussion of the previous sections, could be changed to try new closures, i.e. the SWRC and the hydraulic conductivities in the case of the $R^2$ equation, and the thermal capacity and thermal conductivity in the case of the energy budget. Adding a new SWRC type or a new conductivity functions should be easy.

D.  The largest set of boundary conditions should be smoothly manageable

    E.  The implementation of equations should be abstract, according to the principle of "programming to interfaces and not to concrete classes", which is the core of contemporary OOP (Gamma et al., 1995). The different equations describing the processes should be implemented within the set of classes by implementing a common interface.

    F.  The implementation of algorithms should not depend on the data formats of inputs and outputs.

Another requirement has to do with the user experience. In fact, solvers of PDEs (Menard et al., 2020) tend to be complex to understand and run when features are added. In particular, the number of inputs grows exponentially when features are added, and the user has to overcome a steep learning curve before being able to use these software packages to appreciate all the cases implemented and their physics.

    G.  To simplify this situation, WHETGEO-1D has to be implemented is such a way that any of the alternative implementa-
tions must come only with their own parameters and variables, and appear to the user as simple as possible, though not too simple.

There is finally a last requirement to consider:

    H.  For computational and research purposes, there will be one,two and three dimensional (1D, 2D, 3D) implementations of the aforementioned equations. Therefore, as much as possible of the code should be shared across these. In particular,
the NCZ and Newton algorithms should be shareable across the various applications.

This requirement implies that the geometry of the domain, as well as the topology, be specified in an abstract manner to cope with the specifics of each dimensionality.

The rest of this section is organised to respond to points from A to H. A is actually responded to in the next section describing where the software can be downloaded and with which open license. Points B and C are accomplished by studying
an appropriate design of classes and the use of design patterns (Gamma et al., 1995; Freeman et al., 2008). For D, generic programming is used and specific classes are implemented. E is resolved by deploying a set of classes that implement a common interface with an extension that allows it to obtain the required functionalities.

To respond to the issues raised in F and G, WHETGEO-1D is implemented as various components of OMS3, as shown below, each one with its own inputs. Therefore, the number of flags to check and the number of unused inputs are reduced
to the minimum required by the solvers and parameterizations that the user chooses. If users want to solve the $R^2$ equation alone, for instance, they can pick out the appropriate component and do not need to know about the inputs and details of the energy budget. The separation into components has two other advantages. First, it eases the testing of a single process against





available analytical solutions and against other models results (Bisht and Riley, 2019). Second, it improves the model structure, facilitating the representation of new processes (Clark et al., 2021).

Point H is solved by deploying new components for the 1D, 2D, and 3D cases. In the following section we mainly deal with points B to E. Before discussing details of some classes, a few general choices have to be reported. Data of any type are stored internally in vectors of doubles, in turn encapsulated in appropriate Java objects. OOP good practice would suggest that an object should be immutable (Bloch, 2001), but we decided that the main classes have to be mutable and allocated once forever as singletons (Gamma et al., 1995; Freeman et al., 2008). This potentially exposes the software to side effects but frees it to

allocate new objects at any time step and decreases the computational burden and memory occupancy generated by deallocating unused obsolete objects at runtime. This approach may be considered a specific design pattern for partial differential equation solvers.

### 3.2 The Software Organisation

The more visible effect of our choices is that we have built various OMS3 components:

– whetgeo1d-1.0-beta

– netcdf-1.0-beta

– closureequation-1.0-beta

– buffer-1.0-beta

– numerical-1.0-beta

Internally, the classes are assembled by using some interfaces and abstract classes, since WHETGEO-1D is coded using the Java language.

In order to improve the re-usability of the Java code we adopted a generic programming approach (Berti, 2000) that consists in decoupling of algorithm implementations from the concrete data representation while preserving efficiency. The generic approach has been balanced with domain-specific ones that can improve the computational efficiency of the software, as is the

case of the previously mentioned Thomas algorithm used in 1-D implementations.

Another requirement regards the division of software classes into three main groups, as the lack of a proper separation between the parameterisation of physical processes and their numerical solutions has been recognised as one of the weak points of existing land surface models (Clark et al., 2015b, 2021). One group describes the mathematical-physical problem, the second one implements the numerical solution (Berti, 2000), and the third one contains the growing group of concrete classes.

The first group contains the SWRC, and hydraulic and thermal conductivity, and it forms a stand-alone library since its content can eventually be reused in the 2D and 3D version of WHETGEO-1D. Similarly, we grouped the classes that solve linear and nonlinear algebraic systems, containing the Thomas, conjugate gradient, and the various Netwon types of algorithms, in a second stand-alone library. The third group of classes gathers the concrete implementations and the variety of OMS3 components that are deployed.




The classes used, and their repository for third parties inspection, are illustrated in the 00_Notebooks referred in section
4 and in the supplemental material. However, there are three pivotal groups of classes that we want to mention here: These
contain the description of the geometry of the integration domain, the closure equation, and the state equation.

### 3.2.1 Computational domain, i.e. the `Geometry` class

One of the key aspects to have in a generic solver regards the management of the grid and, in particular, the definition of its
topology, or how grid elements are connected to each other. In the 1D case the description of the topology is quite simple since
it can be implicitly contained in the vector representation: each element of the vector corresponds to a control volume of the
grid and it is only connected with the elements preceding and following it. It is worth noting that this approach is peculiar to
1D problems and cannot be adopted for the 2D and 3D domains, where, especially when unstructured grids are used, the grid
topology requires a smart implementation of the incidence and adjacency matrices.

For each control volume it is necessary to store its geometrical quantities, their position and dimension, its variables, its
parameter set, and the form of the equation to be solved there, referred to in the following as "equation state". The appropriate
arrangement of information together with the internal design of the classes allows us to create a generic finite volume solver.

### 3.2.2 Closure equations, i.e. the `ClosureEquation` abstract class

The `ClosureEquation` class is shown in the UML diagram of Fig. (3). As explained in the Introduction, one of the core
concepts of modelling water and heat transport in soils is the SWRC. Soil is a multi-phase material, thus knowledge of its
composition is of crucial importance in defining its unsaturated hydraulic conductivity and its thermal properties, specific
internal energy and thermal conductivity.

An abstract class `ClosureEquation` is defined to contain only abstract methods that would be overwritten by the con-
crete classes implementing it. The `ClosureEquation` class essentially defines a new data-type. A closer inspection of
Fig. (3) reveals that the `ClosureEquation` is composed by aggregation with the `Parameter` class, which contains all
the physical parameters of the model. Moreover, the `Parameter` class is implemented by using the Singleton pattern (Free-
man et al., 2008). This design pattern is functional to have only one instance of the `Parameter` class shared by all the
`ClosureEquation` objects and to provide a global point of access to the `Parameter` instance.

The Simple Factory pattern `SoilWaterRetentionCurvesFactory` accomplishes the task of implementing the con-
crete classes. By preferring polymorphism to inheritance and using the Factory pattern (Gamma et al., 1995; Freeman et al.,
2008), the developers can easily include and extend existing code or new formulations or parametrisations of SWRC. Besides,
the Simple Factory fulfils the dependency inversion principle (Eckel, 2003), thus new extensions cannot affect the functioning
of existing code. The same closure equation, for instance a particular SWRC, can be used to compute the soil water volume,
when solving the $R^2$ equation, and the specific enthalpy of the soil, when solving the heat equation.





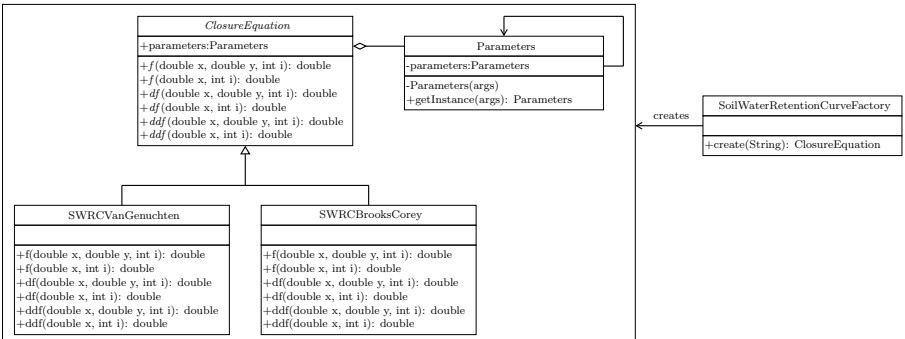

**Figure 3.** UML class diagram for the Java Simple Factory applied for the choice of the SWRC model. The `ClosureEquation` defines the interface that is implemented by the concrete classes `SWRCVanGenucthen` and `SWRCBrooksCorey`. The `ClosureEquation` is aggregated with the class `Parameters` containing the physical parameters of the model. The `Parameters` is implemented by using the Singleton pattern and its instance is inherited by the concrete classes `SWRCVanGenucthen` and `SWRCBrooksCorey`.

### 3.2.3   Equation state, i.e. the `EquationState` class


The `EquationState` class in Fig. (4) contains the implementation of the discretised form of the equation state of the PDE under scrutiny. It contains a reference each to the `ClosureEquation` object, to the `Geometry` and `ProblemVariables` objects.

Notably, the solver of a PDE problem can refer to the abstract class and to its abstract object without implying the specific
concrete equation to be solved or its concrete parameterisations. Moreover, the compositionality of the `EquationState` allows the creation of new solvers from existing closures without the need to add new subclasses. As shown in the UML of Figure 4, the `EquationState` class defines methods used to linearise the PDE when it is nonlinear. For instance, it computes the first and second derivatives, and the functions necessary to define the Jordan decomposition as required by the NCZ algorithm. Specifically, these methods are `p`, `q`, `pIntegral`, `qIntegral`, and `computeXStar`.

In our code design the `ClosureEquation` class is limited to computing a physical parameterisation, whereas the `EquationState` class is used to discretise the equation state of the PDE, and whenever required to properly linearise it. Any new concrete `EquationState` subclass can either have the same physics of another with a different solver or a different physics with the same solver.

### 3.3   Generic Programming at work

As explained in Sect. 2.1.2, the definition of the surface boundary condition for the $R^2$ equation can require the introduction of an additional computation node at the soil surface to simulate the water depth. This means that we have two different equation states, one for the soil water content and one for the water depth. On the other hand, when considering the Neumann or Dirichlet boundary condition we have only one equation state, for the soil water content. The nonlinear solver of the NCZ algorithm must work seamlessly with any type of boundary condition used, and with any type of equation involved in the problem.




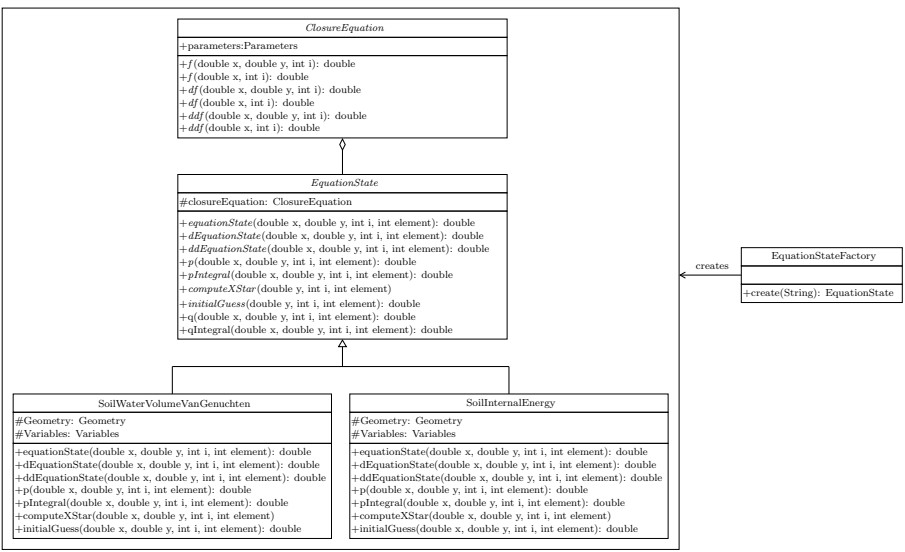

**Figure 4.** UML class diagram for the Java Simple Factory applied for the choice of the `EquationState` model. The `EquationState` defines the interface that is implemented by the concrete classes `SoilWaterVolumeVanGenucthen` and `SoilInternalEnergy`. The `EquationState` contains a reference to a `ClosureEquation` object.

Let us consider, for example, the case of the $R^2$ equation with water depth as the surface boundary condition. Figure (5) reports the pseudocode for computing the nonlinear functions in the NCZ algorithm using a procedural approach. When traversing the computational grid we need to resort to an `if-else` statement to compute the nonlinear function of each control volume: the control volume $N$ represents the control volume for the surface water and in this case we need to use the water depth equation state, $H(\psi)$ in Figure (5). The remaining control volumes represent the soil moisture and for them

we need to use the SWRC equation state, $\theta(\psi)$ in Figure (5). The main limit of this approach is that the computation of the nonlinear functions, $H(\psi)$ and $\theta(\psi)$, is hard-wired into the code of the NCZ algorithm. This presents a shortcoming for the reusability of the code since, as the boundary condition changes and the physical problem changes, it is necessary to modify the `for` loop and the name of the objects computing the nonlinear functions.

     Adopting the OOP and generic programming approach, Fig. (6), it is possible to implement the NCZ algorithm in such a

way that enhances its reusability. The key feature is the decoupling of the computational grid from the algorithm(data) (Berti, 2000). This is achieved through two elements. The first consists in creating a container of the objects that deal with the equation states of the problem, `equationState`, eS in Fig. (6). Second, we use a label `equationStateID`, eSID in Fig. (6), to specify the behaviour of each control volume. So, the behaviour of each control volume is determined by this label and not by the position of the element in the grid. Specifically, when we traverse the grid we use the `equationStateID` to determine

which object inside the container `equationState` to use.

     The NCZ algorithm has been implemented in the `NestedNewtonThomas` class. The `NestedNewtonThomas` contains a reference to the `Thomas` object, whose task is to solve a linear system, and to a list of `EquationState` objects, Fig. (7).





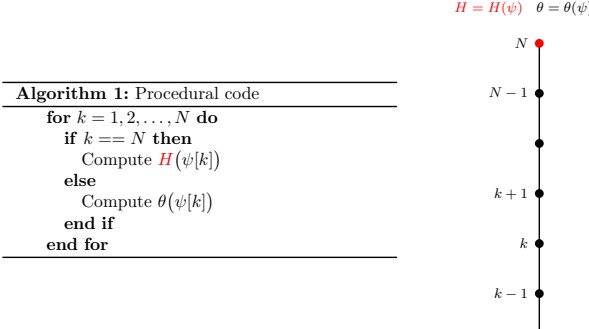

**Figure 5.** Adopting a procedural approach, the computation of the equation states is hard-wired into the code. The behaviour of each control volume is determined by an `if-else` statement according to the position of the element in the grid. In this case the properties of the grid, here the equation state, are joined with the topology. Here the non linear function $V(\psi)$ is replaced with either $H(\psi)$ or $\theta(\psi)$ according with the position of the node. To keep the pseudocode short, $H(\psi)$ and $\theta(\psi)$ stand for all the nonlinear function used in the NCZ algorithm, and the method `f` stands for one of the methods defined in the `EquationState` class.

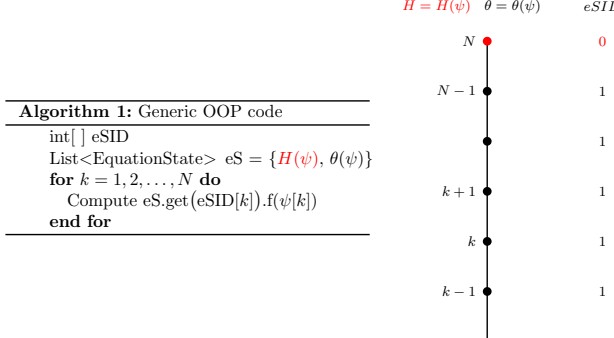

**Figure 6.** Adopting OOP with a generic programming approach, the computation of the equation states is independent from the grid. In fact, the behaviour of each control volume is determined by the vector eSID - `equationStateID` in the code - that determines which object belonging to the class `EquationState` of eS - `equationState` in the code - must be used to compute the equation state. In this manner it is possible to traverse the computation domain without resorting to the `if-else` statement. Here the non linear function $V(\psi)$ is consistently replaced with either $H(\psi)$ or $\theta(\psi)$ according to the position of the node. To keep the pseudocode short, $H(\psi)$ and $\theta(\psi)$ stand for all the nonlinear function used in the NCZ algorithm, and the method `f` stands for one of the methods defined in the class `EquationState`.

Considering the ubiquity of nonlinear problems in Hydrology and the robustness of the NCZ algorithm, the NCZ algorithm has been encapsulated in a stand-alone library.

The example has been illustrated in 1D but it becomes even more effective when working on 2D or 3D, especially with an unstructured grid.



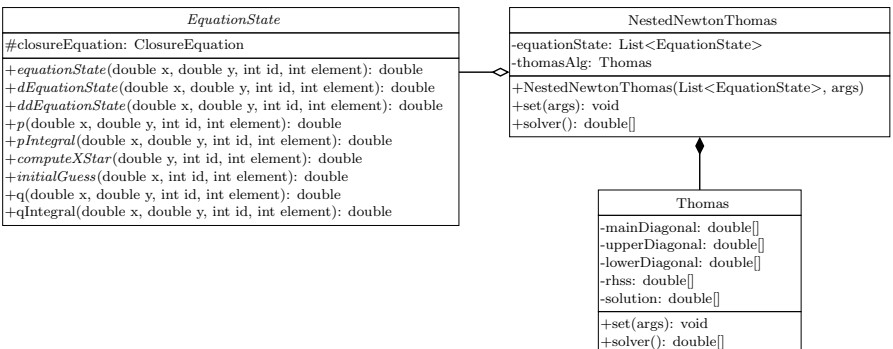

**Figure 7.** UML class diagram of the `NestedNewtonThomas` class. This class deals with the solution of the nonlinear system. The `NestedNewtonThomas` contains a reference each to the `Thomas` object, whose task it is to solve a linear system, and to a list of `EquationState` objects.

## 4   Information for Users and Developers

While most of the what written so far is of general application, the deployment shown here is 1D. Information on WHETGEO-1D for users and developers is provided in the supplemental material, where there is a Jupyter Notebook that contains the

guidelines for executing the codes for any of the components. Its name starts with "00_" and we call it "Notebook Zero" of the components. The latest executable code can be downloaded from

–   https://github.com/geoframecomponents/WHETGEO-1D

and can be compiled by following the instructions therein. The version of the OMS3 compiled project can be found here https://github.com/GEOframeOMSProjects/OMS_Project_WHETGEO1D. The code can be executed in the OMS3 console,

which can be downloaded and installed according to the instructions given at:

–   https://geoframe.blogspot.com/2020/01/the-winter-school-on-geoframe-system-is.html

Some brief information about GEOframe can be found in Appendix B, and more comprehensive information is at:

–   https://abouthydrology.blogspot.com/2015/03/jgrass-newage-essentials.html

–   https://geoframe.blogspot.com/2020/01/gsw2020-photos-and-material.html

To run the tests, please follow the instructions on the Github repository of the GEOframe components. If a user wants to compile the code themselves, they can use the appropriate Gradle script that guarantees independence from any IDE. For further information about input and output formats for WHETGEO-1D, please see the Notebook 00_WHETGEO1D_Richards.ipynb in the folder Documentation of the Zenodo distribution.





### 4.1 Workflow for Users

Examples of uses of WHETGEO-1D can be found in the form of Python Notebooks in the directory Notebooks/Jupyter. Documentation can be found in form of Python Notebooks in the directory Documentation. Simulations with WHETGEO-1D are run as OMS3 simulations. Therefore, the first operation to accomplish is to prepare the appropriate .sim files. For new users, many simulation files are available in the directory simulation of the Zenodo distribution. As shown in Fig. (8), in the modelling solutions that involve WHETGEO-1D, there is always a "Main" component that is in charge of running the core

code for solving the PDE. The inputs and the outputs are treated by other OMS3 components. They are tied together by a Domain Specific Language (DSL) based on Groovy. This allows for great flexibility in using various input and output formats.

### 4.2 Inputs and Outputs

Input data can be broadly classified into time series, computational grid data, and simulation parameters. Time series are used to specify the boundary conditions of the problem. Time series are contained in .csv files with a specific format that is OMS3

compliant (David). With computational grid data we refer to the domain discretisation, initial condition, and soil parameters. All these data are stored in a netCDF file. Time series and computational grid data are elaborated with dedicated Python modules distributed under the gf-group package (Tubini and Rigon, 2021). The simulation parameters, such as the start date and end date of the simulation, time step size, and file paths, are specified by the user in the OMS3 .sim file.

For the design of the output workflow we took advantage of the OMS3 system that allows the user to connect stand-alone

components. Figure (8) shows the output workflow for saving output data and where `Main`, `Buffer`, and `netCDF writer` are the stand-alone OMS3 components. `Main` stands for the generic component having the responsibility of solving the PDE. `Buffer` has the responsibility of temporarily storing output data, and `Writer` handles the saving of data to the disk. The `Buffer` component has the sole purpose of storing data and this has two important advantages. The first is that it limits the number of accesses to the disk to save output, i.e. reducing the computational time. The second is that it introduces a

layer separating the `Main` component from the `netCDF writer`. This increases the flexibility of the modelling solution, as future developers can adopt different file formats, or develop different writer components that, instead of saving all the outputs, can save discrete outputs or aggregated outputs. The advantage is that developers need only know the legacy of the `Buffer` component and customise the both output file format and memory optimisation strategy, such as chuncking, according to their need. Currently all outputs are stored in a netCDF-3 format (Unidata, 2021). netCDF is a self-describing portable data

format developed and maintained by UCAR Unidata. netCDF is commonly used by the Geo-science community and there is an ever-growing number of tools for processing and visualisation.

The choice of including the `Buffer` component in the workflow of the modelling solution is motivated by earlier experiences of the GEOtop community with the GEOtop model, and by more recent experience with the FreeThaw1D model (Tubini et al., 2020), where long spin-ups of the model ($\sim$ 1500 years) were required with consequent large output files ($\sim$ GB). Fur-

thermore, in anticipation of the 2D and 3D developments, the netCDF-3 format is probably not the most appropriate and could be abandoned in favour of more performing file formats Unidata (a, b), such as netCDF-4 or HDF5.





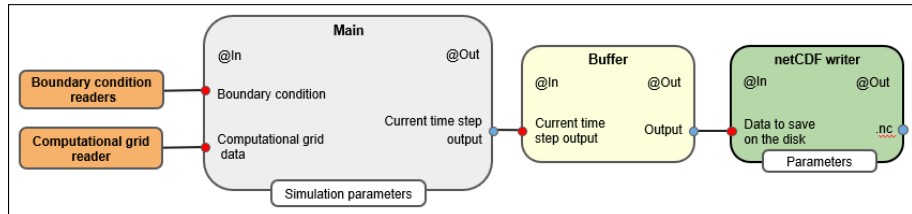

**Figure 8.** Workflow of WHETGEO-1D. The boundary condition readers, computational grid reader, `Main`, `Buffer`, and `netCDF writer` are stand-alone OMS3 components. `Main` stands for the generic component with the responsibility of solving the PDE, the `Buffer` temporarily stores output data that later are passed to the `netCDF writer` component, which in turn saves data to the disk. The `Buffer` component has the sole purpose of storing data. This has two adavantages: the first one is to limit the number of accesses to the disk to save output, i.e. reducing the computational time, and the second one is to introduce a layer separating the `Main` component, which handles the numerical solution of the PDE, and the component responsible for saving outputs.

WHETGEO-1D can be integrated with the built-in calibration component `LUCA` (Hay et al., 2006; Formetta et al., 2014a) and the `Verification` component, as shown in Fig. (9). The former is used to calibrate optimal parameters, the latter to compute the indices of goodness of simulated data versus measured data. Besides the `LUCA` component and the `Verification` component, it is necessary to add two more components, specifically the `Buffer calibration parameters` and the `Measurement point data`. The `Buffer` calibration parameters is needed to interface the `Main` component with the `LUCA` component. In fact, in the WHETGEO-1D `Main` component, physical parameters are stored as vectors, whilst `LUCA` handles calibration parameters as scalars (single value). The `Buffer calibration` component receives the optimal parameters set from the `LUCA` component and returns them packed in appropriate vectors. The `Verification` component receives as input two, OMS3-compliant time series: one for measured data and one for simulated data. In this case it is necessary to extract from the simulation output only the simulated data at the measurement points of the variable used to calibrate the model. These data are then saved as OMS3 time series. It interesting that the integration of WHETGEO-1D with the OMS3 built-in calibration components is achieved by adding two new, stand-alone components without modifying the source code of the existing components, i.e. the `Main` component, the `Buffer` component, and the `netCDF writer`.

## 4.3 Workflow for Developers

Here, as an example, we present how to add the Brooks-Corey (Brooks and Corey, 1964) model as an extension of the code base. The constitutive relationships are given by:

$$\theta(\psi) = \begin{cases} \theta_r + (\theta_s - \theta_r)\left(\dfrac{\psi_d}{\psi}\right)^n & \text{if } \psi \leq \psi_d \\ \theta_s & \text{if } \psi > \psi_d \end{cases} \tag{38}$$

$$K(\psi) = K_s \left[\frac{\theta(\psi) - \theta_r}{\theta_s - \theta_r}\right]^{3+\frac{2}{n}} \tag{39}$$



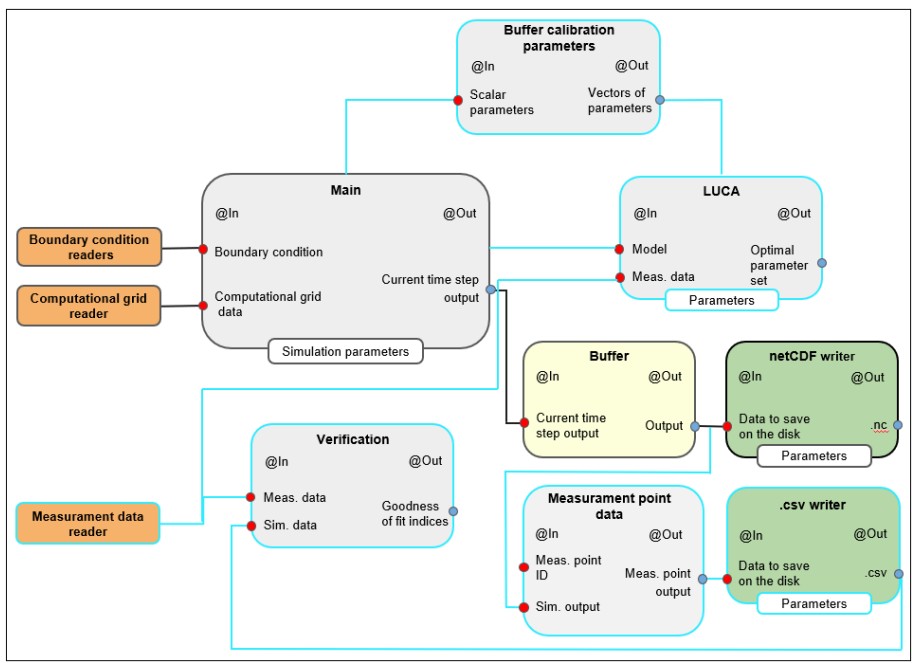

**Figure 9.** Workflow of WHETGEO-1D integrated with LUCA. The cyan lines identify those components required to integrate WHETGEO-1D with LUCA. The LUCA and Verification components are built-in OMS3 components. The former is used to calibrate optimal parameters, the latter to compute the indices of goodness of simulated data versus measured data. To integrate WHETGEO-1D with the OMS3 calibration components it is necessary to add two other components into the workflow, specifically the Buffer calibration parameters component and the measurement point data component. LUCA handles scalar parameters whilst WHETGEO 1D uses vectorial parameters. The `Buffer calibration parameters` creates an interface between these two components, simply creating vectorial parameters from scalar parameters. The `Verification` component requires the input time series as a .csv file of simulated data at the measurement points. To accomplish this requirement, it is necessary to modify the output strategy: the `Buffer` component passes output data to the `Measurement points data` component. This component extracts only the simulated variables at the measurement points and passes them to the OMS3 time series writer, `.csv writer`, which save the simulated time series as a .csv file. Notably, if the user is not interested in saving all the output of each run of the model during calibration, it is possible to remove the `netCDF writer` component.

where $\theta_s$ and $\theta_r$ are, respectively, the saturated and residual values of the volumetric water content, $\psi_d$ is the air-entry water suction value, $n$ is the pore size distribution index, and $K_s$ is the saturated hydraulic conductivity at saturation.

The standard approach to adding a new SWRC parametrization, here the Brooks and Corey model, requires the definition a new class that extends the abstract class `ClosureEquation`. This new class, `SWRCBrooksCorey`, provides the implementation of the abstract methods defined in the super class `ClosureEquation`, and inherits the association with the Parameters class. Specifically, the `SWRCBrooksCorey` class overrides the following methods:

– `f` calculates the water content for a given water suction value and set of parameters, Eq. (38);





    – `df` calculates the first derivative of Eq. (38);

    – `ddf` calculates the second derivative of Eq. (38).

In order to use the Brooks-Corey model in the Richards' equation it is necessary to define a new class, `SoilWaterVolumeBrooksCorey`, that extends the abstract class `StateEquation`. Specifically the `SoilWaterVolumeBrooksCorey` class overrides the following methods:

    – `equationState` calculates the water volume using the Brook-Corey model for a given water suction value and set of parameters, Eq. (18).

– `dEquationState` calculates the first derivative of the `equationState` function, in this the moisture capacity function. This method is used within the linearisation algorithm.

    – `ddEquationState` calculates the second derivative of the `equationState` function. This method is relevant for those models where the $\psi^*$ cannot be computed analytically but requires the application of a root finding method such as the bisection method. An example is the soil internal energy function when considering the phase change of water
Tubini et al. (2020).

    – `p` calculates the $p$ function of the Jordan decomposition.

    – `pIntegral` calculates the $V_1$ function of the Jordan decomposition.

    – `computeXStar` calculates the $\psi^*$ value to properly define the functions $p$ and $V_1$.

    – `initialGuess` calculates the initial guess for the linearisation algorithm.

**5 Conclusions**

In this paper we discussed the issues raised by implementing a new expandable system to model the Earth's CZ. The issues faced in doing so were grouped in various concerns:

    – physical-hydrological;

    – mathematical;

– software engineering.

The implementation has been shown to solve the issues presented in 7 observations, 3 requirements, and A to H design specifications. Each of these was analysed and informed the choice of algorithms and code implementation. The first deployment of the concepts was the 1D stand-alone water budget and the coupled water and energy budgets WHETGEO versions.

    The water budget was tested against analytical solutions presented in (Srivastava and Yeh, 1991) and (Vanderborght et al.,
2005). Some behavioural simulations were also performed to show some features of the code, such as the ability to deal with





switching boundary conditions. Also, we showed that WHETGEO can be easily extended to simulate the thermal regime in frozen soils, as described in Tubini et al. (2020), by merely adding the SFC model presented by (Dall'Amico et al., 2011). As described in the "Code Availability" section, WHETGEO-1D is an open source software distributed with the GPL v3.0 license. Its development is pursued with Open Source tools available on Github. Also, its documentation was produced using only open

source tools and is available along with the distribution of the main code.

*Code availability.* The source code is written in Java using the object-oriented programming paradigm. The source code can be found at https://github.com/geoframecomponents/WHETGEO-1D and a frozen version at 10.5281/zenodo.5112727. The OMS3 project can be found at https://github.com/GEOframeOMSProjects/OMS_Project_WHETGEO1D and a frozen version is available on Zenodo at 10.5281/zenodo. 5112727. The source code of external dependencies are provided at in the README of the github page https://github.com/geoframecomponents/

WHETGEO-1D. WHETGEO-1D is deployed as an open source code to work alone or within the Object Modelling System version 3 framework (David et al., 2013). In the latter case it can be connected at run time with the many other components developed in the GEOframe system to provide hydrometeorological forcings and other fluxes, such as evapotranspiration. The simulations presented here can be found at 10.5281/zenodo.4749319. The code must be run within the OMS3 Console or using the Dockerized version of OMS3. To set up the environment please follow the steps described in the README file present in the Github repository https://github.com/GEOframeOMSProjects/

OMS_Project_WHETGEO1D and in the GEOframe pages at https://geoframe.blogspot.com/2021/05/whetgeo-1d.html (last accessed 18 May 2021). Once you have installed OMS3, please follow the instructions contained in the Documentation folder. They contain all the details about simulations inputs and parameters.

## Appendix A: GEOframe

WHETGEO-1D was implemented as a Java component within the GEOframe, an open-source, component-based hydrological

modelling system. Within GEOframe, each part of the hydrological cycle is implemented in a self-contained building block, an OMS3 component (David et al., 2013). Components can be joined together to obtain multiple modelling solutions that can accomplish from simple to very complicated tasks. GEOframe has proved great flexibility and robustness in several applications (Bancheri et al., 2020; Abera et al., 2017a, b). There are more than 50 components available that can be grouped into the following categories:

– Geomorphic and DEM analyses;

– Spatial extrapolation/interpolation of meteorological variables;

– Estimation of the radiation budget;

– Estimation of evapotranspiration;

– Estimation of runoff production with integral distributed models;

– Channel routing;





- Travel time analysis;

- Calibration algorithms.

Using the components for geomorphic and DEM analyses Rigon et al. (2006b), the basin can be discretised into Hydrological Response Units (HRUs), i.e., hydrologically similar parts, such as a catchment or a hillslope or one of its parts. The meteoro-
logical forcing data can be spatially interpolated using a geostatistical approach, such as the Kriging technique (Bancheri et al., 2018). Both shortwave and longwave radiation components are available for the estimation of the radiation budget (Formetta et al., 2013, 2016). Evapotranspiration can be estimated using three different formulations: the FAO Evapotranspiration model (Allen et al., 1998), the Priestley-Taylor model (Priestley and Taylor, 1972), and the Prospero model, (Bottazzi, 2020; Bottazzi et al., 2021). Snow melting and the snow water equivalent can also be simulated with three models, as described in (Formetta et al., 2014b). Runoff production is performed by using the Embedded Reservoir Model (ERM) or a combination of its reservoirs (Bancheri et al., 2020). The discharge generated at each hillslope is routed to the outlet using the Muskingum-Cunge method (Bancheri et al., 2020). Travel time analysis of a generic pollutant within the catchment can be done using the approach proposed in (Rigon et al., 2016b, a). Model parameters can be calibrated using two algorithms, and several objective functions: Let Us CAlibrate (LUCA) (Hay et al., 2006) and Particle Swarm Optimization (PSO) (Kennedy and Eberhart, 1995). A graph-based structure, called NET3 (Serafin, 2019), is employed for the management of process simulations. NET3 is designed using a river network/graph structure analogy, where each HRU is a node of the graph, and the channel links are the connections between the nodes. In any NET3 node, a different modelling solution can be implemented and nodes (HRUs or channels) can be connected or disconnected at run-time through scripting. GEOframe is open source and helps the reproducibility and replicability of research (Bancheri, 2017). Developers and users can easily collaborate, share documentation, and archive examples and data within the GEOframe community.

## Appendix B: OMS3

The Object Modeling System v.3 (OMS3) is a component-based environmental modelling framework that provides a consistent and efficient way to: 1) create science simulation components; 2) develop, parameterise, and evaluate environmental models, and modify and adjust them as science advances; and 3) re-purpose environmental models for emerging customer requirements (David et al., 2013).

In OMS3 the term component refers to self-contained, separate software units that implement independent functions in a context-independent manner (David et al., 2013). This means that developers and researches can build their model as composition of stand-alone components, moving away from the monolithic approach. The entire GEOframe system, and therefore WHETGEO-1D, is built upon the OMS3 framework.

Compared to other Environmental Modelling Frameworks (EMF), OMS3 is characterised by being a non-invasive and lightweight framework (Lloyd et al., 2011). That is to say, the model code is not tightly coupled with the underlying framework - OMS3 -, i.e. the environmental modeller does not need a deep knowledge of the API, and the modelling components can still





function and continue to evolve outside the framework (David et al., 2013). In fact, OMS3 relies on specific annotations to provide meta data for Java code. These annotations describe elements such as classes, fields, and methods, and are used by the framework to interpret the component as a building block of the modelling solution (MS), hence controlling its connectivity and data flow (David et al., 2013). It is worth noting that, being meta data, these annotations do not directly affect the execution of the source code outside the OMS3 - non-invasive and lightweight framework.

Besides the technical aspects, the adoption of a software framework has a positive effect on "non-functional" quality attributes, such as maintainability, portability, reusability and understandability (David et al., 2013). The component-based approach allows the developer to break down the problem into smaller parts, each one tackled by a specific component. Hence, the components are joined together to build the desired modelling solution (point B). This facilitates the construction of new MSs, thanks to the plug-in system of model components (David et al., 2013; Peckham et al., 2013; Serafin, 2019). Thanks to the modularity, the updating of a component with the most recent scientific advances is facilitated and has no side effects on the other components. The other advantage regards the long term development of the code. From past experiences, one of the main limits to model development and maintenance was related to the lack of a proper software architectural design (Rizzoli et al., 2006; David et al., 2013; Formetta et al., 2014a; Bancheri, 2017; Serafin, 2019). Moreover, it is interesting to note that the component-based approach encourages collective model development (Serafin, 2019) and also eases the attribution of authorship since any component is a stand-alone chunk of code and can be authored separately.

Besides, the adoption of an environmental modelling framework promotes the concept of reproducible research, easing third parties inspection and providing consistent and verifiable model results (Formetta et al., 2013; Bancheri, 2017; Serafin, 2019).

Another advantage of using OMS3 is represented by the opportunity to keep the code development transparent to the user.

## Appendix C: $R^2$ test cases

In this section we test the solver of the $R^2$ equation against the analytical solutions presented by Srivastava and Yeh (1991) and by Vanderborght et al. (2005). Then we present and discuss two "behavioural" test cases to try out WHETGEO-1D in simulating both the infiltration excess and the saturation excess process.

### C1  Analytical solution Srivastava and Yeh (1991)

Srivastava and Yeh (1991) derived an analytical solution describing the one-dimensional transient infiltration in an homogeneous and layered soil. The hydraulic properties of the soil are described by the following constitutive relations:

$$K(\psi) = K_s e^{\alpha \psi} \tag{C1}$$

$$\theta(\psi) = \theta_r + (\theta_s - \theta_r) e^{\alpha \psi} \tag{C2}$$

where $Ks$ is the saturated hydraulic conductivity, $\theta_r$ is the residual water content, $\theta_s$ is the saturated water content, and $\alpha$ is the soil pore-size distribution parameter, representing the desaturation rate of the SWRC. The lower boundary condition

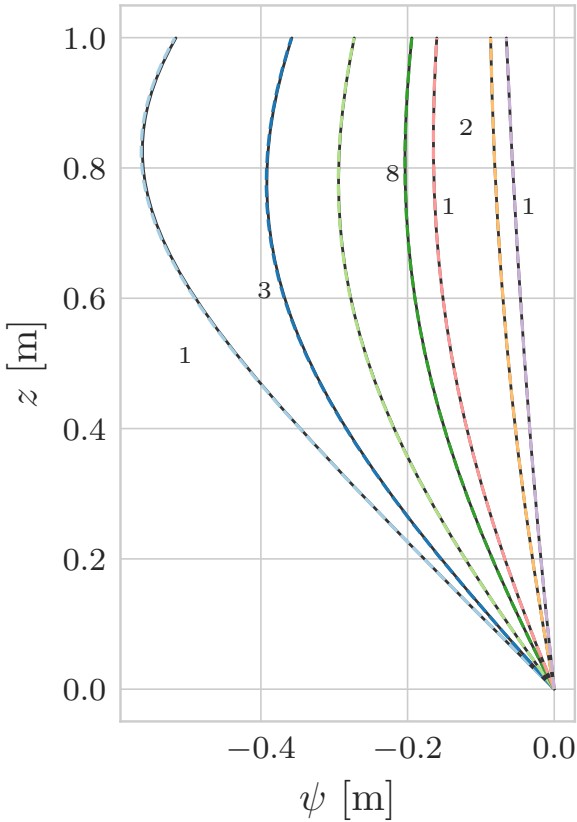

**Figure C1.** Comparison between the analytical and numerical solutions for the test problem TP1.

is represented by the water table, $\psi = 0$, while the upper boundary condition is subjected to a constant flux, $q$. The initial

condition corresponds to the steady state profile due to a prescribed initial flux at the soil surface and prescribed pressure at the lower boundary. The analytical solution is derived by linearising Richards' equation and using Laplace's transformation. Details on the analytical solution can be found in (Srivastava and Yeh, 1991).

### C1.1    Homogeneous soil

We consider a one-dimension homogeneous soil layer of 1 [m] depth (TP1). The saturated hydraulic conductivity value is

assumed to be 1.0 [cm/h], with $\theta_s = 0.45$, $\theta_r = 0.2$, and $\alpha = 0.01$ [1/cm]. The initial condition is determined by imposing as lower boundary condition $\psi = 0$ and a constant water flux at the soil surface $q_A = 0.1$ [cm/h]. For times greater than 0 the water flux at the soil surface is $q_B = 0.9$ [cm/h]. The domain is discretised with a uniform grid space $\Delta z = 0.001$ [m] and the time step is $\Delta t = 60$ [s]. The model accuracy is enhanced by allowing two Picard iterations per time step. Figure (C1) shows a comparison between the numerical and the analytical solutions.



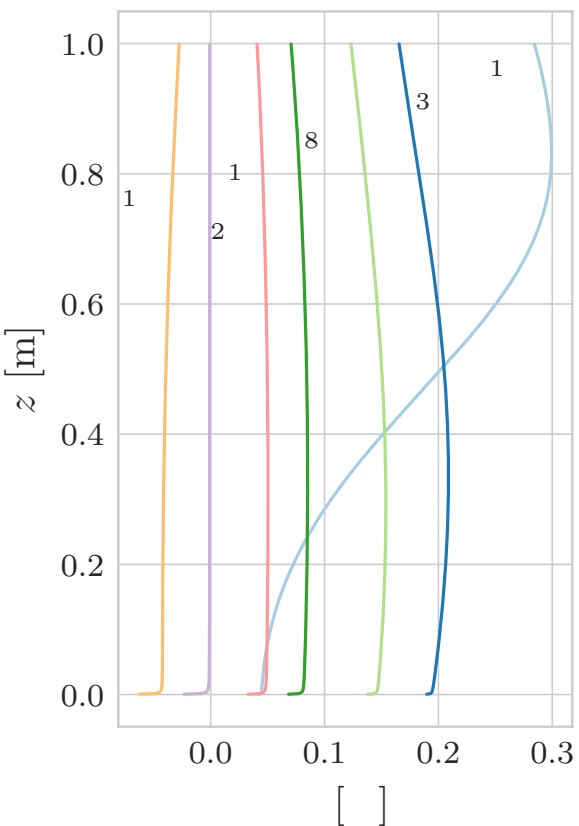

**Figure C2.** Relative water suction error for the test problem TP1.

## C1.2 Layered soil

In this numerical problem (TP2) we consider one-dimensional vertical infiltration toward the water table through a layered soil. The initial condition is determined by imposing as lower boundary condition $\psi = 0$ and a constant water flux at the soil surface $q_A = 0.1$ [cm/h]. For times greater than 0 the water flux at the soil surface is $q_B = 0.9$ [cm/h]. The domain is discretised with a uniform grid space $\Delta z = 0.001$ [m] and the time step is $\Delta t = 60$ [s]. The model accuracy is enhanced by allowing two Picard iterations per time step. The hydraulic conductivity at the interface is computed as the harmonic mean of the neighbours (Romano et al., 1998). Comparison between the numerical and the analytical solution for water suction is shown in Fig. (C3).

## C2 Analytical solution Vanderborght et al. (2005)

The next test case was defined by Vanderborght et al. (2005) to evaluate the steady-state flux in layered soil profiles. For this numerical problem (TP3) we consider a soil column of 2 [m] depth with one soil type for depth 0 [m] − 0.5 [m] overlying another soil type for depth 0.5 [m] − 2 [m], specifically for loam over sand, sand over loam, and clay over sand. The soil





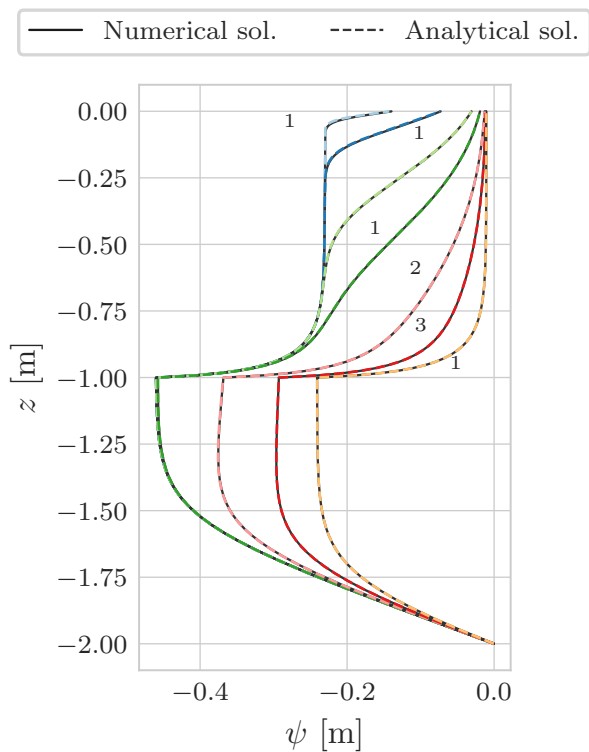

**Figure C3.** Comparison between the analytical and numerical solutions for the test problem TP2.

**Table C1.** Hydraulic properties for the three soil types used in the Vanderborght test case (TP3).

| Soil type | $\theta_s$ [-] | $\theta_r$ [-] | $\alpha$ [m$^{-1}$] | $n$ [-] | $K_s$ [m s$^{-1}$] |
|---|---|---|---|---|---|
| Sand | 0.43 | 0.045 | 15.0 | 3 | $1.16e-04$ |
| Loam | 0.43 | 0.08 | 4.0 | 1.6 | $5.79e-06$ |
| Clay | 0.4 | 0.1 | 1.0 | 1.1 | $1.16e-06$ |

parameters are defined in Tab. (C1). The initial condition for water suction is a uniform profile with $\psi = -20$ [m], the surface boundary condition is a constant flux $q = 5.79\,10^{-8}$ [m s$^{-1}$], and at the bottom we impose a free drainage boundary condition. The domain is discretized with a uniform grid space $\Delta z = 0.01$ [m] and the time step is $\Delta t = 3600$ [s]. In order to reach the steady state condition the simulation lasts 2 years. Comparison between the numerical and the analytical solution is shown in
Fig. (C5).



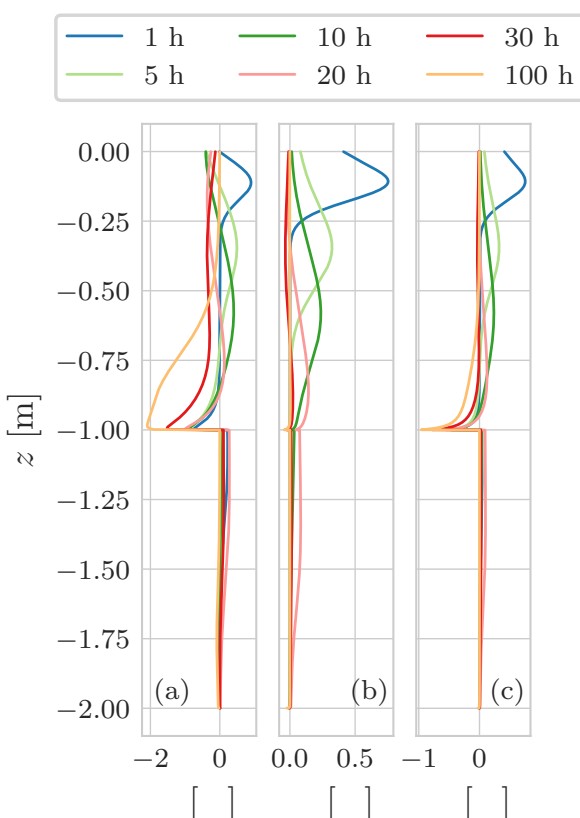

**Figure C4.** Comparison of relative water suction error $\delta$ for the test problem TP2 using different interface hydraulic conductivity algorithms. Panel (a) is computed with max., panel (b) with harmonic mean, and panel (c) with geometric mean. As reported in (Romano et al., 1998), the harmonic mean offers the best agreement with the analytical solution. This is particularly evident at the interface between the two layers.

## C3 Surface boundary condition

The definition of the surface boundary condition is a nontrivial task since it is a system-dependent boundary condition. The infiltration rate through the soil surface depends on precipitation, rainfall intensity $J$, and on the moisture condition of the soil. Because of this, the surface boundary condition may change from the Dirichlet type - prescribed water suction - to the Neumann type - prescribed flux - and vice-versa. The works by Horton (1933) and Dunne and Black (1970) establish the conceptual framework to explain the runoff generation.

The infiltration excess or Horton runoff occurs when the rainfall intensity is larger than infiltration capacity of the soil:

$$\left| J \right| > \left| - K(\psi) \frac{\partial}{\partial z} (\psi + z) \right|_{z=0} \tag{C3}$$

Infiltration excess is most commonly observed with short-duration, intense rainfall.



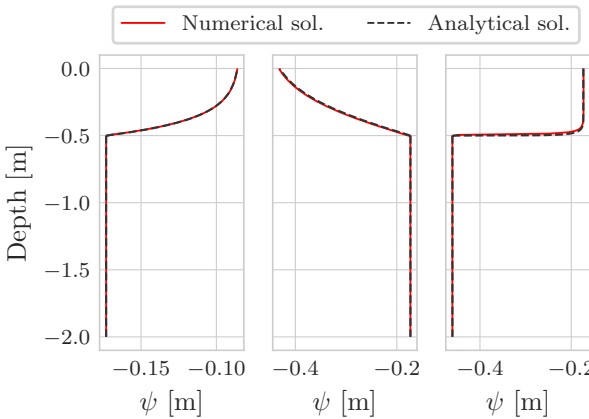

**Figure C5.** Comparison between the analytical and the numerical solution for the test problem TP3. The three panels show the vertical profile of water suction at steady state for a constant flow rate of $5.79\,10^{-8}$ [m s$^{-1}$]: (a) clay-sand soil profile, (b) loam-sand soil profile, and (c) sand-loam soil profile.

**Table C2.** Hydraulic properties of the silty clay loam soil Bonan (2019) for the Horton runoff numerical experiment.

| $\theta_r$ m$^3$m$^{-3}$ | $\theta_s$ m$^3$m$^{-3}$ | $\alpha$ m$^{-1}$ | $n -$ | $K_s$ ms$^{-1}$ |
|---|---|---|---|---|
| 0.089 | 0.43 | 1.0 | 1.23 | $1.9447e-07$ |

The saturation excess or Dunnian runoff occurs when the soil is saturated and additional water exfiltrates at the soil surface. Saturation excess generally occurs with long-duration, moderate rainfall, or with a series of successive precipitation events. In this case the soil depth or the presence of shallow fragipan are determining factors for saturation excess. Another possible cause is the rise of the water table up to the soil surface.

### C3.1    Infiltration excess

In this numerical experiment we consider a homogeneous soil of $3$ m depth. Soil hydraulic properties are described with the Van Genuchten's model, Table (C2).

The initial condition is assumed to be hydrostatic with $\psi = 0$ m at the bottom. The surface boundary condition is a synthetic rainfall, as in Fig. (C6) (a), lasting $15$ min with constant intensity of $0.028$ mms$^{-1}$. At the bottom we prescribed a Dirichlet boundary condition with constant $\psi = 0$ [m] so the transient is driven only by the surface boundary condition. In Fig. (C6)

panel (a), the time is indicated when it would be necessary to switch from the Neumann type to the Dirichlet type boundary condition.

Figure (C7) shows a comparison of water ponding at the soil surface considering two different initial conditions of the soil, wet and dry. For the wet case, the initial condition is hydrostatic with $\psi = 0$ m at the bottom. For the dry case, the initial condition is hydrostatic with $\psi = -100$ m at the bottom. In the wet initial condition the hydraulic conductivity is higher than

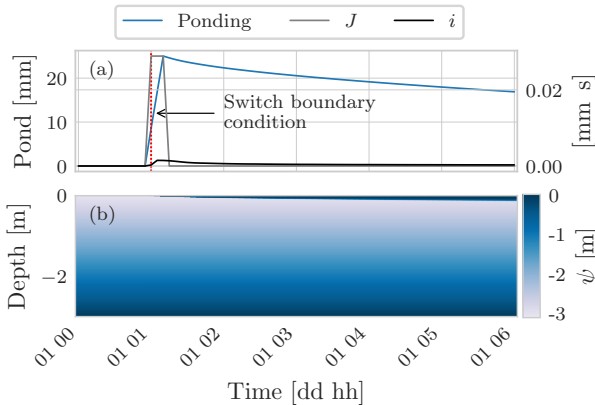

**Figure C6.** Panel (a) shows a comparison between the rainfall intensity $J$ and actual soil infiltration $i$. The rainfall intensity exceeds the actual infiltration rate so water builds up at the soil surface (blue line). Panel (b) shows the time evolution of the water suction within the soil. From the numerical point of view, as water builds up at the soil surface it would be necessary to switch the boundary condition, from Neumann type to Dirichlet type.

**Table C3.** Hydraulic properties of the loam layer and clay layer, respectively, Bonan (2019) for the numerical experiment on Dunnian runoff due to water table rising.

| $\theta_r$ m$^3$m$^{-3}$ | $\theta_s$ m$^3$m$^{-3}$ | $\alpha$ m$^{-1}$ | $n\,-$ | $K_s$ ms$^{-1}$ |
|---|---|---|---|---|
| 0.078 | 0.43 | 3.6 | 1.56 | $2.8889e-06$ |
| 0.068 | 0.38 | 0.8 | 1.09 | $5.5556e-07$ |

for the dry initial condition, however, in the dry case the capillary gradient is larger and because of this the soil infiltration capacity is higher, as in Fig. (C7) panel (a). With regards to the water ponding, the maximum value is almost the same in both cases, 1 mm higher in the wet case, but the time evolution is different: in the wet case the water only infiltrates completely 13 h later than the dry case. This delay may seem counter-intuitive since wetter conditions are associated with higher values of hydraulic conductivity, Fig. (C8), but in the wet soil the capillary gradients are smaller than in the dry soil, Fig. (C9).

**C3.2 Saturation excess**

In this section we present two numerical experiments to simulate the saturation excess process. Saturation excess is more critical, in terms of simulation stability, than the infiltration excess (Forums). We consider two cases: one in which the water table reaches the soil surface; and another in which the total rainfall amount is larger than the maximum water holding capacity but the rainfall intensity is less than the maximum infiltration rate.

Firstly, we consider a layered soil of 3 m depth. The thicknesses of the loamy layer and clay layer are, respectively, 0.5 m, and 2.5 m. The soil hydraulic properties are described with the Van Genuchten's model, Tab. (C3). The initial condition is



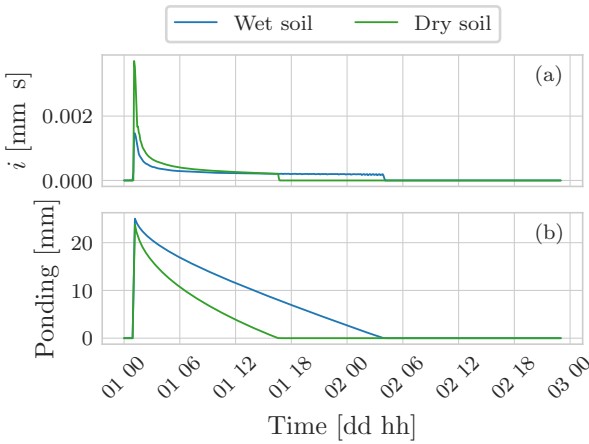

**Figure C7.** Panel (a) shows a comparison between the infiltration rate for two cases: wet and dry initial condition. In the dry case, soil infiltration is greater than the wet case even though the hydraulic conductivity is smaller. This is due to the higher capillary gradients that develop in the soil. Panel (b) shows the time evolution of the water ponding at the soil surface.

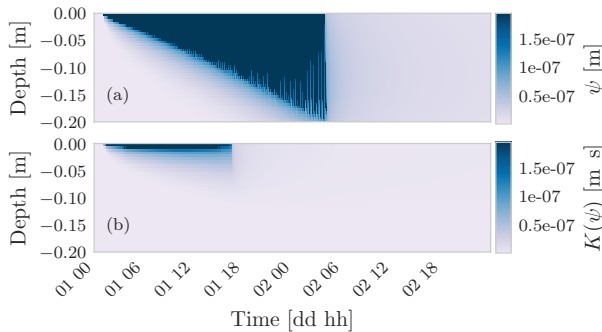

**Figure C8.** Panel (a) shows the hydraulic conductivity field for the case of wet soil, while panel (b) shows the hydraulic conductivity field for the case of dry soil.

assumed to be hydrostatic with $\psi = 0$ m at the bottom. At the surface boundary condition we prescribe no rainfall, while at the bottom a variable Dirichlet boundary condition is prescribed Fig. (C10) (a). The transient is driven by variation of the water table. In Fig. (C10) panel (a) the time is indicated when it would be necessary to switch the surface boundary condition from the Neumann type to the Dirichlet type boundary condition and vice-versa.

Secondly, we consider a layered soil of 3 m depth. The thicknesses of the loamy sand layer and the clay layer are, respectively, 0.3 m, and 2.7 m. The soil hydraulic properties are described with the Van Genuchten's model, Tab. (C4). The initial condition is assumed to be hydrostatic with $\psi = -2$ m at the bottom. The surface boundary condition is a synthetic rainfall Fig. (C11) (a), at the bottom we prescribed a Dirichlet boundary condition with constant $\psi = -2$ m so the transient is driven


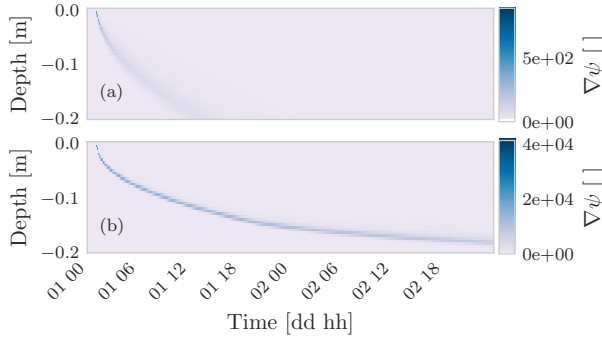

**Figure C9.** Panel (a) shows the capillary gradient for the case of wet soil, panel (b) shows the the capillary gradient for the case of dry soil. As can be seen, in the dry soil the capillary gradient is two orders of magnitude larger than in the wet soil. Because of this higher gradient water infiltrates faster in the dry soil than in the wet soil.

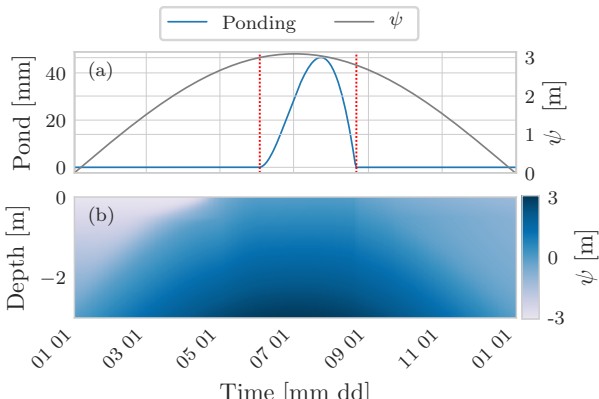

**Figure C10.** Panel (a) shows the water table position, Dirichlet boundary condition, and the water ponding at the soil surface. The dotted red lines indicate the times at which the boundary condition at the soil surface should be switched. The first line indicates the switch from the Neumann type to the Dirichlet type since water starts building up. The second line indicates the switch form the Dirichlet type to the Neumann type because there is no water at the soil surface. Panel (b) shows the time evolution of the water suction within the soil.

only by the surface boundary condition. Figure (C11) panel (b) shows the time evolution of the degree of saturation within the soil. Initially water infiltrates in the soil but then the clay layer, which is characterised by a lower conductivity than the loam-sand layer, limits the deep infiltration causing the saturation of the loam-sand layer from below.

Repeating the above numerical experiment with a thicker loam-sandy layer, Fig. (C12), there is no water ponding at the soil surface. In this case all the rainfall can infiltrate into the loam-sandy layer thanks to the increased water storage capacity.





**Table C4.** Hydraulic properties of the loamy sand layer and clay layer, respectively, Bonan (2019) for the numerical experiment on Dunnian runoff due rainfall.

| $\theta_r$ m$^3$m$^{-3}$ | $\theta_s$ m$^3$m$^{-3}$ | $\alpha$ m$^{-1}$ | $n-$ | $K_s$ ms$^{-1}$ |
|---|---|---|---|---|
| 0.057 | 0.41 | 12.4 | 2.28 | $4.0528e-05$ |
| 0.068 | 0.38 | 0.8 | 1.09 | $5.5556e-07$ |

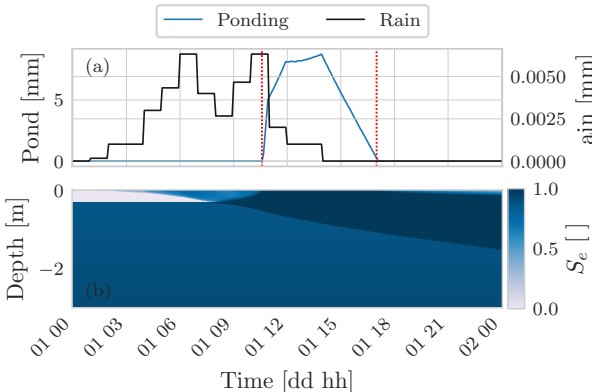

**Figure C11.** Panel (a) shows the rainfall and the water ponding at the soil surface. Initially rainfall can infiltrate into the soil and no water builds up a the soil surface. As the uppermost layer of the soil saturates water starts ponding at the soil surface. The dotted red lines indicate the times at which the boundary condition at the soil surface should be switched. Panel (b) shows the time evolution of the saturation degree within the soil.

## Appendix D: Energy budget

Here we present a behavioural test case on the pure heat conduction considering the surface energy balance. The purpose of this simulation is twofold. First, we show an application of WHETGEO-1D that exploits existing GEOframe components to model the external components of the surface energy budget, specifically: the incoming shortwave radiation (Formetta et al., 2013), the incoming longwave radiation (Formetta et al., 2016), and the latent heat flux. Second, we show how to easily include the phase change of water by adding a new closure equation that describes the SFC model presented by Dall'Amico et al. (2011)

The soil column is 30 m deep and the initial condition is a constant temperature profile $T = 12\,°C$. Figure (D1) panel (a) shows the components of the surface energy fluxes and the thermal regime of the uppermost 2 m of the soil column. As can be seen in panel (b), the soil temperature falls below $0\,°C$; therefore it is not reasonable to neglect freezing and thawing processes.

By including the closure equation of the SFC presented by Dall'Amico et al. (2011), it is possible to consider the phase change of water. Moreover, compared to the FreeThaw-1D model Tubini et al. (2020), in WHETGEO-1D it is possible to drive the simulation of the soil thermal regime by using the surface energy budget. This aspect represent a novelty with respect to

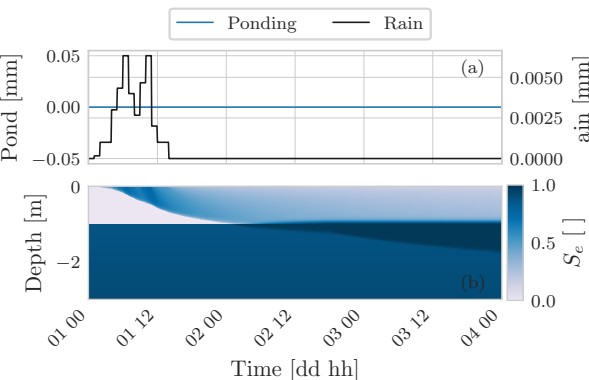

**Figure C12.** In this numerical experiment there is no saturation excess since the greater thickness of the uppermost layer ensure a sufficient water storage capacity for the forcing rainfall.

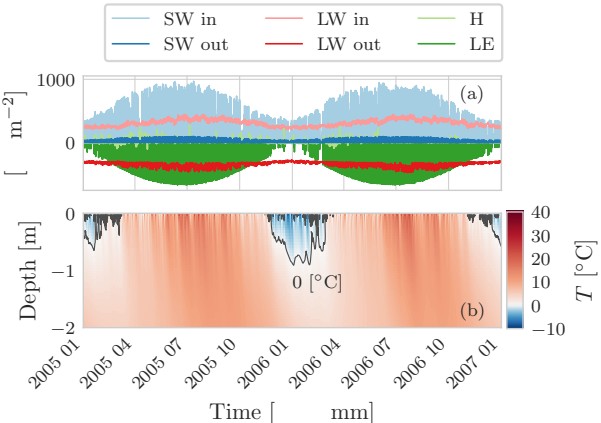

**Figure D1.** Behavioural test case of pure heat conduction in soil considering the surface energy budget. Panel (a) shows the surface energy fluxes driving the simulation. The external fluxes, incoming shortwave and longwave radiation, and the latent heat flux, are computed with existing GEOframe components. Panel (b) shows the thermal regime of the uppermost 2 m of the soil column. As can be seen, during winter the temperature of the uppermost layer goes below 0 °C, the grey line is the 0 °C isotherm, therefore, it is not reasonable to overlook the phase change of water.

FreeThaw-1D and was obtained with a minimum effort thanks to the code design we adopted. The soil column is 30 m depth and the initial condition is a constant temperature profile $T = 12$ °C. The parameters of the SFC are presented in Tab. (D1)

Figure (D1) shows in panel (a) the components of the surface energy fluxes, and the thermal regime of the uppermost 2 m of the soil column. As can be seen in panel (b), the soil temperature fall below the 0 °C thus neglecting freezing and thawing processing results in a strong approximation.

Figure (D3) shows a comparison between the two simulations.



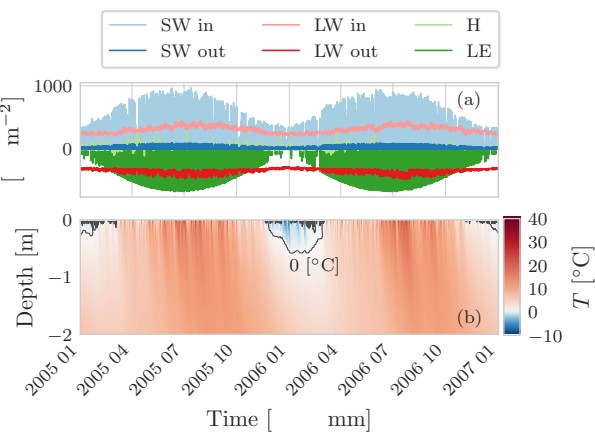

**Figure D2.** Behavioural test case of the pure heat conduction in soil considering the surface energy budget and the phase change of water. Panel (a) shows the surface energy fluxes driving the simulation. The external fluxes, incoming shortwave and longwave radiation, and the latent heat flux, are computed with existing GEOframe components. Panel (b) shows the thermal regime of the uppermost 2 m of the soil column.

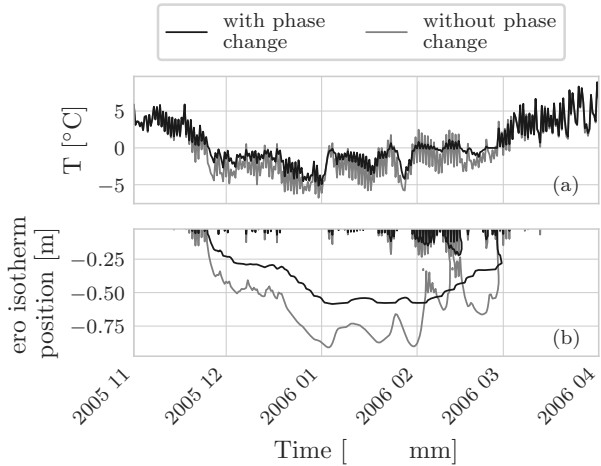

**Figure D3.** Panel (a) shows a comparison of the soil temperature at 0.05 m considering the phase change of water and without. As can be seen, in the latter case the soil temperature reaches lower value and fluctuates more: when considering the phase change of water we include in the problem the latent heat of water that increase the thermal inertia of the soil. Panel (b) shows a comparison of the position of the zero-isotherm in the two simulation. When the phase of water is not included the zero-isotherm does deeper into the ground.



**Table D1.** Parameters of the SFC model.

| $\theta_r \ \mathrm{m^3 m^{-3}}$ | $\theta_s \ \mathrm{m^3 m^{-3}}$ | $\alpha \ \mathrm{m^{-1}}$ | $n\ -$ |
|---|---|---|---|
| 0.068 | 0.38 | 0.8 | 1.09 |

*Author contributions.* Conceptualization, R.R. and N.T.; Methodology: N.T. and R.R. Software engineering: N.T. and R.R.; Code writing
N.T. Code Revision: RR and NT; Data curation and simulations, N.T.; Paper Writing, Review, Editing; N.T. and R.R.; Language revision
Documentation: N.T., R.R. and Concetta D'Amato; Funding: R.R.

*Competing interests.* The authors declare no conflict of interest.

*Acknowledgements.* The authors would like to thank Professor Vincenzo Casulli and Professor Michael Dumbser of the Department of Civil,
Environmental and Mechanics Engineering at the University of Trento for their fruitful discussions on the numerical aspects of the work. We
thank Concetta D'Amato for the help in preparing the supporting material. We would like to thank Joseph Eamonn Tomasi for his invaluable
help with the linguistic revision. This work has been partially supported by a Ph.D. grant by the Department of Civil, Environmental and
Mechanics Engineering at the University of Trento and by the Italian MIUR Project (PRIN 2017) "WATer mixing in the critical ZONe:
observations and predictions under environmental changes-WATZON" (project code: 2017SL7ABC).





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
