# Peer review of "Implementing the Water, HEat and Transport model in GEOframe WHETGEO-1D v.1.0: algorithms, informatics, design patterns, open science features, and 1D deployment."

_Geoscientific Model Development, 2021_

## Author Comment (AC1)

Dear Anonymous Referee #1,

Thank you very much for your review and your constructive comments. The entire text of your comment is shown (**RC**) together with our authors' responses (**AR**).

Kind regards,

Niccolò Tubini and Riccardo Rigon

**RC**: This paper dose a very good job of describing the design idea, software engineering, and the technical issues about implementing a physically based model - WHETGEO, to simulate the water and energy budgets in a soil column. All the text is easy to follow and understandable. The figures, tables, and equations are well prepared and organized. The authors gave a good review on the mathematical and numerical issues involved in solving the Richardson-Richards equation.

**AR**: We thank the reviewer for the good review of the paper, we try in the following to answer convincingly to their comments. Hoping to make it easier for the reader to read the answers, this first comment has been divided sentence by sentence.

**RC**: However, what's new in this paper is not easy to find.

**AR**: That is certainly our fault, and we will try to tell it better in the revised manuscript. In summary:

- The model is an entirely new code. It is open-source and built with open-source tools. It and its documentation fulfil the requirements of Open Science, as those presented, for instance, in (Hall, et al. 2021). In a science where all is based on computer simulations, this is a necessary requirement that most of the existing codes do not fulfil. Besides, it is built with a chain open-source tool.

- The model made available for the first time to the public an algorithm of integration, the NCZ, also known as Nested Newton, which has a priori convergence assured for any time step and for a great variety of conditions (which does not happen for other solvers, to our knowledge)

- The component solving the Richardson-Richards equation allows for accumulating water on top of the column without resorting to switching boundary condition or other techniques, as done in current software like Hydrus-1D, by simply solving a unique system of equations.

- We think that the code design is a step forward with respect to present implementations of $R^2$ codes in term of application of object-oriented paradigms whose advantages, we believe we have clearly explained in the text and try to convey better in the revised manuscript. These advantages, are:

- o Easy expandability of the existing Soil Water Retention Curves (SWRC) and Hydraulic Conductivity (HC). As an example, we have described in Section 4.3 how to add the Brooks and Corey SWRC model.

- o The possibility to use different SWRC and HC schemes in different points of the domain (this could be considered weird but is a possibility that can be useful in some use case).

    The code allows for safe transition between surface water (see point 2) and groundwater equations see, (for understanding the issue see, for instance, Staudinger et al. 2019).

- • WHETGEO comes within an existing system, GEOframe, that allows its connection with other water budget compartments whose modelling can be pursued independently but synergistically, as codes for the interception (over the surface), the evaporation from soil and the transpiration. These last topics cannot be covered in the present paper whose scope has the limited ambition to present the WHETGEO structure and design.

**RC**: Some ideas and proposals mentioned in the front part are not well addressed in the latter. The purpose of incorporating heat transport and surface energy budget and their advantages are not well demonstrated.

**AR**: The reviewer is right. We did not clarify well that mentioning all the issues related to the physical processes was functional to get the proper software design not to solve all the issues. Instead, we want to propose a system where new scientific ideas can be fitted without having to disrupt the existing codes, i.e., our goal is to permit quantum leap in science in an incremental way (from the code point of view). In the revised manuscript we will try to make it clear.

**RC**: The overall impression of this paper is about assembling some available models or parameterizations in a new style.

**AR**: Thanks for this comment since we recognize that we did not clearly explain the motivation behind our work, and we discuss about the informatics only in Section 3. In the revised manuscript we partially anticipate the problems related to the informatics in the Introduction.

The rationale behind this work is based on the past experiences in developing scientific software, like GEOtop (Rigon et al., 2006). It is evident how the lack of a proper software design led to difficulties in maintaining existing code and develop it (Serafin 2019, Heaton, D., & Carver, J. C. 2015) . Citing the IDEAS project *'the software productivity is one critical aspect of scientific productivity'* ([https://ideas-productivity.org/ideas-classic/how-to/](https://ideas-productivity.org/ideas-classic/how-to/)), that cannot be overlooked. Therefore, in doing hydrological science today we cannot ignore having good, stable, open-source, well-engineered, inspectionable, and documented software. Our contribution aims to present the software design of WHETGEO and showing its advantages such as: the possibility to easily include the representation of new processes and new parameterization, as well

the possibility to easily change the algorithms to solve the equations. Obviously, we proved that the software is working properly by comparing it with known results.

After all, the Journal name is Geoscientific Model Developments, and we believe to have said much new stuff on how to do such type of models in our paper.

RC: There can be some improvements on the science part and I hope the authors put more attention on the science part instead of the software engineering in writing, especially for the scientific significance and internal logics.

AR: The rebuttal of the previous comments contains already part of the answer. The Journal is not Theory and Processes in Geoscience, but GMD and appropriate recognition should be given to the proper building of the models. Please allow us to disagree with the observation that we do not put enough attention of the science part. We think that the advance of science does not boil down to present new sets of equations or parameterizations for modelling water and energy budget, or other processes, but the advance of science consists also the adoption of new numerical tools made by new research and developing software according to the new need accrued by past experiences. It is worth to note that the paper presenting the nested Newton algorithm (Casulli and Zanolli, 2010) has been published almost 10 years ago, and the latest review papers (Farthing and Ogden, 2017, Zha et al., 2019) still discuss of the numerical limits of current solvers for the Richardson-Richards equation. Besides, most of the issues presented in the recent preprint by Ragenass et al. (2021) are solved with our code, which is, it seems very important for a new generation of Soil-Water-Atmosphere models.

Dilettantism in software (models) building may reflect in biased analysis and results of the hydrological processes threatening the development of science since its ground. We hope that we can be judged on these software methodological aspects even if we did not advance the present knowledge of the processes we are modelling. We provide a tool with which the processes will be investigated reliably without unknown side effect.

Detailed comments

1. RC: In the conclusion part, it reads that "The implementation has been shown to solve the issues presented in 7 observations, 3 requirements, and A to H design specifications". There is no problem with "A to H design specifications". But for the "7 observations, 3 requirements", they are not well "solved", or not easy to find at least, in the current presentation. This is one major concern of this paper.

   AR: The reviewer is right. We did not answer to the to the 7 observations. While we certainly satisfied the 3 requirements. The 7 observations were functional to the code design and not tasks be solved in the present work. Some of them require insights that are still to be gained through properly design experiments.

Moreover, the scope of our present research is to properly build a software that can be easily developed to cope with these observations and not, for instance, perform laboratory or field experiments. To be more specific, we did not answer to the observations 1 to 4. However, the code has some answers also to observations 5 to 7. Mainly:

- The code allows for changing easily soil water retention curves and therefore, for instance, it would be easy to implement new ones taking more accurately account for the different processes in the various ranges of suction. A description on how to add a new soil water retention curve is available in Section 4.3.

- The effect of temperature on viscosity is already in. We will add some supplemental material to the revised manuscript where we show what happens when temperature changes.

- Freezing and thawing processes coupled with Richardson-Richards equation is going to be inserted. Currently only the heat conduction with phase change is in the code. For showing this, we added something about in the supplemental material.

2. **RC**: Following point 1, there are "7 observations, 3 requirements" which have been solved as mentioned. Are they all unique features of the currently proposed model? Since there is no comparison with other existing models.

**AR:** We mentioned above the role of the 7 observations and what we have solved so far. Asking for comparison with existing models is a usual request from reviewers. However, our answer has almost always been that it is not a fair one. The right place for these intercomparisons are papers where each particular model is run by its Authors or researchers well-trained on that model. By the researchers who do the comparison, benchmark use cases are issued, and everybody match the results of the models against the benchmark. This in turn produces "community, shared, papers" whose reliability can still be challenged but represent a fairer picture than a comparison made by a single, biased group of biased researchers. Running several different models assumes that:

- the models are freely available,

- a consistent amount of time is invested in learning the details of the implementation of each model in order to have a fair comparison between models.

- the researchers are unbiased with respect the model.

Actually, if pursued, the comparison pushes away for months the publication of good material threatening especially the younger co-authors of the papers with debatable advancement for science. Because we are not blind, however, we gave a look to issues raised in the Hydrus-1D mailing list. One of the most relevant was mentioned in Answer 2.2 above. Others are:

- problems arise in simulating saturation excess https://www.pc-progress.com/forum/viewtopic.php?f=3&t=3632 for which we have presented a synthetic test case,

- sometimes the convergence is not guaranteed https://www.pc-progress.com/forum/viewtopic.php?f=3&t=692,https://www.pc-progress.com/forum/viewtopic.php?t=2582,https://www.pc-progress.com/forum/viewtopic.php?f=1&t=150, an issue we do not have.

3. **RC**: What's the new development of the model in physical about this paper? Besides the software engineering or technical part. As the paper said in the abstract "a new, physically based…"

**AR:** The equations we solve are well established in literature to capture infiltration processes (form this the adjective "physically based"), but the same is not true for their numerical solution. Thus, one novelty regards the algorithms we used. Evidence of this is the attention we have paid in analysing, referring to existing literature, the pitfalls in solving the Richardson-Richards equation and the energy budget.
For instance, applications such as that presented in (Regenass et al. 2021) could benefit of the improvements in the algorithms we presented in our paper. In WHETGEO the solution of the Richardson-Richards equation does not suffer of convergence problem as the time step or the grid size increase. The adoption of the mixed form of the equation is preferred over the saturation form since it allows to model jointly the saturated and unsaturated conditions. Moreover, the coupling strategy with ponding water makes WHETGEO naturally suitable to investigate the partitioning between infiltration and surface run-off.

The other novelty is on the informatics side. As previously pointed out this an open-source code developed adopting an object-oriented approach and a generic programming paradigm so it can be further developed and customized by other researchers with little effort and minimal knowledge of the code language (Java in our case).
In Appendix D, we show how to incorporate the recent advance presented in (Tubini et al. 2021) with a minimum effort. About the problem of freezing and thawing processes we discussed deeply in point 7.

In our opinion there is no cheating in saying that a model is new if its informatics and its algorithms are absolutely new. There is no need to come out with a new physics as soon as we show that we provide new treatments of the old physics which potentially could bring to new insights when applied to new studies.

Hoping to ease the reader in following the answers, comment 4. has been divided in smaller parts.

**4.1 RC**: If the purpose of this paper is to assemble some models already developed, the test cases in the appendix are very common.

**AR:** What does they means with "already developed"? Do they mean that they know other models from which we took the code design? That we already had the code? Certainly, the test cases in Appendix are know, we have taken from literature. If the reviewer knows some other test cases, we invite them to suggest them to us and we'll use them in the revised manuscript.

**4.2 RC:** More importantly, there is no comparison with other models. So, the advantages are not well demonstrated.

**AR:** We already expressed our opinion about comparisons above.

**4.3 RC:** The resolution of the time step (60 s) and soil grids are all very small.

**AR:** The reviewer is correct, and we could add as supplemental material about the comparison between numerical and analytical solution using coarser grid. However, we would like to point out that small time steps (and fine grids) are required to maintain accuracy (in the sense used in Numerics), and that the adoption of larger time steps and coarser grids, do not affect the convergence of the solver and the conservation of the mass is still guaranteed, while it is more critical by using other algorithms.

**4.4 RC:** For the energy budget and phase change (Appendix D), only the result and difference are presented. The energy budget and phase change are common functions in land surface modelling or hydrological modelling. Because there is nothing special in the case design in the appendix, many other models may also reproduce such a result.

**AR**: The reviewer is right but what change in our model is the numerics used to solve these equations. About the heat advection-diffusion equation, currently the conservative and non-conservative forms are used interchangeably although this choice has sometime large effects on the numerical solution. Moreover, the coupling between the Richardson-Richards equation and the heat advection-diffusion equation is done using the numerical model presented in Casulli and Zanolli (2005) that to our knowledge has not been applied before to solve this problem.

As regards the problem of the heat conduction in presence of phase change, we agree with the reviewer that it is common problem in hydrological modelling, and we are aware that several models are presented in literature cope with it. Again, although this problem is commonly found in several model, we would like to

remark that its solution is far from being trivial and resolved, as discussed in Tubini et al. 2021.

5. **RC**: Eq. (8) (Line 105) is not further mentioned.
   **AR**: This will be corrected in the revised manuscript.

6. **RC**: Please check with (26), (27), (29) and (30) about the sign of operation.
   **AR**: This will be corrected in the revised manuscript

7. **RC**: Some units in Figs. 2, C2, C4, C9, C11, D1, D2, and D3 are missing.
   **AR**: They will be added in the revised version

8. **RC**: Please unify the soil layer index, k or i. Such as in Eq. (21), k in Figure 1, and some other places.
   **AR**: Sorry for the inconvenient, this will be done in the revised manuscript

**References**

Campoy, A., Ducharne, A., Cheruy, F., Hourdin, F., Polcher, J., & Dupont, J. C. (2013). Response of land surface fluxes and precipitation to different soil bottom hydrological conditions in a general circulation model. *Journal of Geophysical Research: Atmospheres*, 118(19), 10-725.

Casulli, V., & Zanolli, P. (2005). High resolution methods for multidimensional advection–diffusion problems in free-surface hydrodynamics. *Ocean Modelling*, 10(1-2), 137-151.

Farthing, M. W., & Ogden, F. L. (2017). Numerical solution of Richards' equation: A review of advances and challenges. *Soil Science Society of America Journal*, 81(6), 1257-1269.

Hall, C. A., Saia, S. M., Popp, A. L., Dogulu, N., Schymanski, S. J., Drost, N., ... & Hut, R. (2021). A Hydrologist's Guide to Open Science. Hydrology and Earth System Sciences Discussions, 1-23. https://doi.org/10.5194/hess-2021-392

Heaton, D., & Carver, J. C. (2015). Claims about the use of software engineering practices in science: A systematic literature review. *Information and Software Technology*, 67, 207-219.

Regenass, D., Schlemmer, L., Jahr, E., & Schär, C. (2021). It rains and then? Numerical challenges with the 1D Richards equation in kilometer-resolution land surface modelling. *Hydrology and Earth System Sciences Discussions*, 1-33.

Rigon, R., Bertoldi, G., & Over, T. M. (2006). GEOtop: A distributed hydrological model with coupled water and energy budgets. *Journal of Hydrometeorology*, 7(3), 371-388.

Serafin, F. (2019). Enabling modeling framework with surrogate modeling capabilities and complex networks (Doctoral dissertation, University of Trento).

Staudinger, M., Stoelzle, M., Cochand, F., Seibert, J., Weiler, M., & Hunkeler, D. (2019). Your work is my boundary condition!: Challenges and approaches for a closer collaboration between hydrologists and hydrogeologists. *Journal of Hydrology*, 571, 235-243.

Tubini, N., Gruber, S., & Rigon, R. (2021). A method for solving heat transfer with phase change in ice or soil that allows for large time steps while guaranteeing energy conservation. *The Cryosphere*, 15(6), 2541-2568.

Zha, Y., Yang, J., Zeng, J., Tso, C. H. M., Zeng, W., & Shi, L. (2019). Review of numerical solution of Richardson–Richards equation for variably saturated flow in soils. Wiley Interdisciplinary Reviews: *Water*, 6(5), e1364.

---

## Author Comment (AC2)

Dear Anonymous Referee #2,

Thank you very much for your review and your constructive comments. The entire text of your comment is shown (RC) together with our authors' responses (AR).

Kind regards,

Niccolò Tubini and Riccardo Rigon

**RC**: The paper is well written, and it address the important topic of implementing Richards equation in a land surface code. It is useful to have both radiation estimation in complex terrain and the surface energy implemented.

I'm not really sure of the utility of changing water viscosity in infiltration processes, in fact the diurnal cycle in soil field measurements are always to be watched carefully because of temperature dependent erros in instruments (especially electronics). However, solving the heat transport equation is useful for the surface energy budget evaluation.

I think that it can be published after it is modified following the remarks of reviewer 1.

**AR**: We thank the reviewer for the good review of the paper. About the dependence of water viscosity on temperature, thus also of the saturated hydraulic conductivity, we are aware of the possible errors in measurement and that measurement must be interpreted carefully but we think that a numerical tool should offer the possibility to the user to also include this type of modelling solution.

It is possible that other factors overshadow temperature effect on saturated hydraulic conductivity (Lentz, 2001) but it is worth to point out that in arid regions, and bare soil, especially if dark and containing peat, may experience high temperature in summer. In this case temperature can become a leading factor in controlling the infiltration rate, therefore it is advisable that numerical simulators can account for the temperature effect on saturated hydraulic conductivity (Vereecken et al., 2019).

On this opportunity there are two aspects that we think should be considered:

- In the foreseeable future the instruments will be better than today's so we will be more confident with the measurements.
- From the informatic point of view, i.e. the possibility to further develop WHETGEO-1D, the dependence of the saturated hydraulic conductivity has been included in such a way that not only it is possible to change the formulation relating temperature and water viscosity but allows the developers to easily include other factors that modify the saturated hydraulic conductivity, for instance the dependence on depth (Decharme et al. 2006, Shlemmert et al. 2019).

References

Decharme, B., Douville, H., Boone, A., Habets, F., & Noilhan, J. (2006). Impact of an exponential profile of saturated hydraulic conductivity within the ISBA LSM: simulations over the Rhône basin. Journal of Hydrometeorology, 7(1), 61-80.

Lentz, R. D., & Bjorneberg, D. L. (2002). Influence of irrigation water properties on furrow infiltration: Temperature effects. Sustaining the Global Farm.  Selected papers from the 10th International Soil Conservation Organization Meeting held May 24-29, (1999) at Purdue University and the USDA-ARS National Soil Erosion Research Laboratory.

Schlemmer, L., Schär, C., Lüthi, D., & Strebel, L. (2018). A groundwater and runoff formulation for weather and climate models. Journal of Advances in Modeling Earth Systems, 10(8), 1809-1832.

Vereecken, H., Weihermüller, L., Assouline, S., Šimůnek, J., Verhoef, A., Herbst, M., ... & Xue, Y. (2019). Infiltration from the pedon to global grid scales: An overview and outlook for land surface modelling. Vadose Zone Journal, 18(1).

---

## Author Response (AR1)

Dear Editor,

We appreciate the clear language of the two Anonymous Referees and the opportunity to further clarify and revise the manuscript.

Besides the changes in the manuscript, we have added a supplement containing the comparison against the analytical solution presented by Srivastava and Yeh 1991 using different time step size and spatial discretizations, and the infiltration excess and saturation excess simulations considering the effect of temperature on the saturated hydraulic conductivity.

Since our work aims to present a new software, we added as asset the material of the recent GEOframe Summer School we held on WHETGEO-1D. The material we shared consists of the slides of lectures as well as the recorded videos, and the tutorials to use WHETGEO-1D. This is meant to make it easier to use and disseminate WHETGEO-1D.

Kind regards,
Niccolò Tubini and Riccardo Rigon

**List of relevant changes**

- Line 101-102 we change the unit.
- We added Line 111 to clarify the meaning of $T_1$ and $T_2$.
- We added Section 1.5 *Informatics* where we explain the problems related to the informatics. This section anticipates and extends what was written in Lines 374-384 of the submitted manuscript accordingly with the comments of the Anonymous Referee #1 comments. Please note that in new version Lines 374-384 of the submitted manuscript has been removed. Consequently Section 3 has been renamed as *Design and deployment of WHETGEO-1D*.
- Section 1.6 we clarified that the 3 requirements and the 7 observations are necessary to properly ground the software design. Moreover, here we specified that one of the novelties of WHETGEO regards the adoption of new numerical tools in order to overcome some numerical iusses of current solvers. The changes in Section 1.6 answer to the observations of the Anonymous Referee #1, specifically *Detailed comments 1*.
- Line 150-151 we reformulated the sentence.
- In Section 2.2.2 we fixed the signs of the equations.
- In the Conclusions we summed up the novelties within WHETGEO specifying that we coped with all the 3 requirements, and we solved only observation 1-4, whereas 5-7 are not fully addressed.
- Figures 1, 5, 6 we changed index $k$ to $i$.
- Figure 2 we fixed the y-label in panel (b).
- Section C1.1 C1.2 we added the unit to physical quantities.
- Figure C2 we fixed the unit in the x-label.
- Figure C3 we fixed the y coordinate values.
- Table C1 we fixed the units in the heading.
- Figure C4 we fixed the unit in the x-label and the y coordinate values.
- Figure C6, C7, C8, C9, C11 we fixed the units in the y-labels.
- Figure D1 we fixed the units in the x-labels.